# Deep physical neural networks trained with backpropagation

Logan G. Wright[1,2,4✉], Tatsuhiro Onodera[1,2,4✉], Martin M. Stein[1], Tianyu Wang[1], Darren T. Schachter[3], Zoey Hu[1] & Peter L. McMahon[1✉]

Deep-learning models have become pervasive tools in science and engineering. However, their energy requirements now increasingly limit their scalability[1]. Deep-learning accelerators[2–9] aim to perform deep learning energy-efficiently, usually targeting the inference phase and often by exploiting physical substrates beyond conventional electronics. Approaches so far[10–22] have been unable to apply the backpropagation algorithm to train unconventional novel hardware in situ. The advantages of backpropagation have made it the de facto training method for large-scale neural networks, so this deficiency constitutes a major impediment. Here we introduce a hybrid in situ–in silico algorithm, called physics-aware training, that applies backpropagation to train controllable physical systems. Just as deep learning realizes computations with deep neural networks made from layers of mathematical functions, our approach allows us to train deep physical neural networks made from layers of controllable physical systems, even when the physical layers lack any mathematical isomorphism to conventional artificial neural network layers. To demonstrate the universality of our approach, we train diverse physical neural networks based on optics, mechanics and electronics to experimentally perform audio and image classification tasks. Physics-aware training combines the scalability of backpropagation with the automatic mitigation of imperfections and noise achievable with in situ algorithms. Physical neural networks have the potential to perform machine learning faster and more energy-efficiently than conventional electronic processors and, more broadly, can endow physical systems with automatically designed physical functionalities, for example, for robotics[23–26], materials[27–29] and smart sensors[30–32].

Like many historical developments in artificial intelligence[33,34], the widespread adoption of deep neural networks (DNNs) was enabled in part by synergistic hardware. In 2012, building on earlier works, Krizhevsky et al. showed that the backpropagation algorithm could be efficiently executed with graphics-processing units to train large DNNs[35] for image classification. Since 2012, the computational requirements of DNN models have grown rapidly, outpacing Moore's law[1]. Now, DNNs are increasingly limited by hardware energy efficiency.

The emerging DNN energy problem has inspired special-purpose hardware: DNN 'accelerators'[2–8], most of which are based on direct mathematical isomorphism between the hardware physics and the mathematical operations in DNNs (Fig. 1a, b). Several accelerator proposals use physical systems beyond conventional electronics[8], such as optics[9] and analogue electronic crossbar arrays[3,4,12]. Most devices target the inference phase of deep learning, which accounts for up to 90% of the energy costs of deep learning in commercial deployments[1], although, increasingly, devices are also addressing the training phase (for example, ref. [7]).

However, implementing trained mathematical transformations by designing hardware for strict, operation-by-operation mathematical isomorphism is not the only way to perform efficient machine learning. Instead, we can train the hardware's physical transformations directly to perform desired computations. Here we call this approach physical neural networks (PNNs) to emphasize that physical processes, rather than mathematical operations, are trained. This distinction is not merely semantic: by breaking the traditional software–hardware division, PNNs provide the possibility to opportunistically construct neural network hardware from virtually any controllable physical system(s). As anyone who has simulated the evolution of complex physical systems appreciates, physical transformations are often faster and consume less energy than their digital emulations. This suggests that PNNs, which can harness these physical transformations most directly, may be able to perform certain computations far more efficiently than conventional paradigms, and thus provide a route to more scalable, energy-efficient and faster machine learning.

[1]School of Applied and Engineering Physics, Cornell University, Ithaca, NY, USA. [2]NTT Physics and Informatics Laboratories, NTT Research, Inc., Sunnyvale, CA, USA. [3]School of Electrical and Computer Engineering, Cornell University, Ithaca, NY, USA. [4]These authors contributed equally: Logan G. Wright, Tatsuhiro Onodera. ✉e-mail: lgw32@cornell.edu; to232@cornell.edu; pmcmahon@cornell.edu

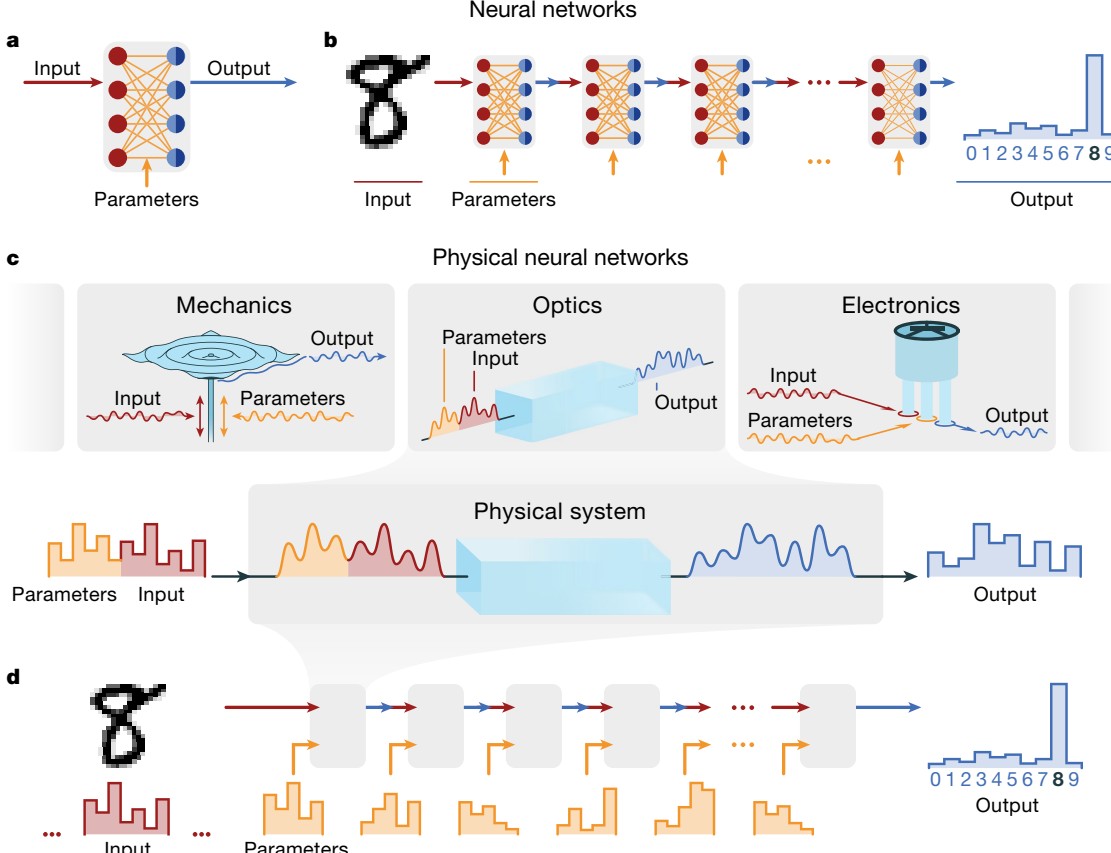

Neural networks

Physical neural networks

**Fig. 1 | Introduction to PNNs. a**, Artificial neural networks contain operational units (layers): typically, trainable matrix-vector multiplications followed by element-wise nonlinear activation functions. **b**, DNNs use a sequence of layers and can be trained to implement multi-step (hierarchical) transformations on input data. **c**, When physical systems evolve, they perform, in effect, computations. We partition their controllable properties into input data and control parameters. Changing parameters alters the transformation performed on data. We consider three examples. In a mechanical (electronic) system, input data and parameters are encoded into time-dependent forces (voltages) applied to a metal plate (nonlinear circuit). The controlled

multimode oscillations (transient voltages) are then measured by a microphone (oscilloscope). In a nonlinear optical system, pulses pass through a $\chi^{(2)}$ crystal, producing nonlinearly mixed outputs. Input data and parameters are encoded in the input pulses' spectra, and outputs are obtained from the frequency-doubled pulses' spectra. **d**, Like DNNs constructed from sequences of trainable nonlinear mathematical functions, we construct deep PNNs with sequences of trainable physical transformations. In PNNs, each physical layer implements a controllable physical function, which does need to be mathematically isomorphic to a conventional DNN layer.

PNNs are particularly well motivated for DNN-like calculations, much more so than for digital logic or even other forms of analogue computation. As expected from their robust processing of natural data, DNNs and physical processes share numerous structural similarities, such as hierarchy, approximate symmetries, noise, redundancy and nonlinearity[36]. As physical systems evolve, they perform transformations that are effectively equivalent to approximations, variants and/or combinations of the mathematical operations commonly used in DNNs, such as convolutions, nonlinearities and matrix-vector multiplications. Thus, using sequences of controlled physical transformations (Fig. 1c), we can realize trainable, hierarchical physical computations, that is, deep PNNs (Fig. 1d).

Although the paradigm of constructing computers by directly training physical transformations has ancestry in evolved computing materials[18], it is today emerging in various fields, including optics[14,15,17,20], spintronic nano-oscillators[10,37], nanoelectronic devices[13,19] and small-scale quantum computers[38]. A closely related trend is physical reservoir computing (PRC)[21,22], in which the transformations of an untrained physical 'reservoir' are linearly combined by a trainable output layer. Although PRC harnesses generic physical processes for computation, it is unable to realize DNN-like hierarchical computations. In contrast, approaches that train the physical transformations[13–19]

themselves can, in principle, overcome this limitation. To train physical transformations experimentally, researchers have frequently relied on gradient-free learning algorithms[10,18–20]. Gradient-based learning algorithms, such as the backpropagation algorithm, are considered essential for the efficient training and good generalization of large-scale DNNs[39]. Thus, proposals to realize gradient-based training in physical hardware have appeared[40–47]. These inspiring proposals nonetheless make assumptions that exclude many physical systems, such as linearity, dissipation-free evolution or that the system be well described by gradient dynamics. The most general proposals[13–16] overcome such constraints by performing training in silico, that is, learning wholly within numerical simulations. Although the universality of in silico training is empowering, simulations of nonlinear physical systems are rarely accurate enough for models trained in silico to transfer accurately to real devices.

Here we demonstrate a universal framework using backpropagation to directly train arbitrary physical systems to execute DNNs, that is, PNNs. Our approach is enabled by a hybrid in situ–in silico algorithm, called physics-aware training (PAT). PAT allows us to execute the backpropagation algorithm efficiently and accurately on any sequence of physical input–output transformations. We demonstrate the universality of this approach by experimentally performing image

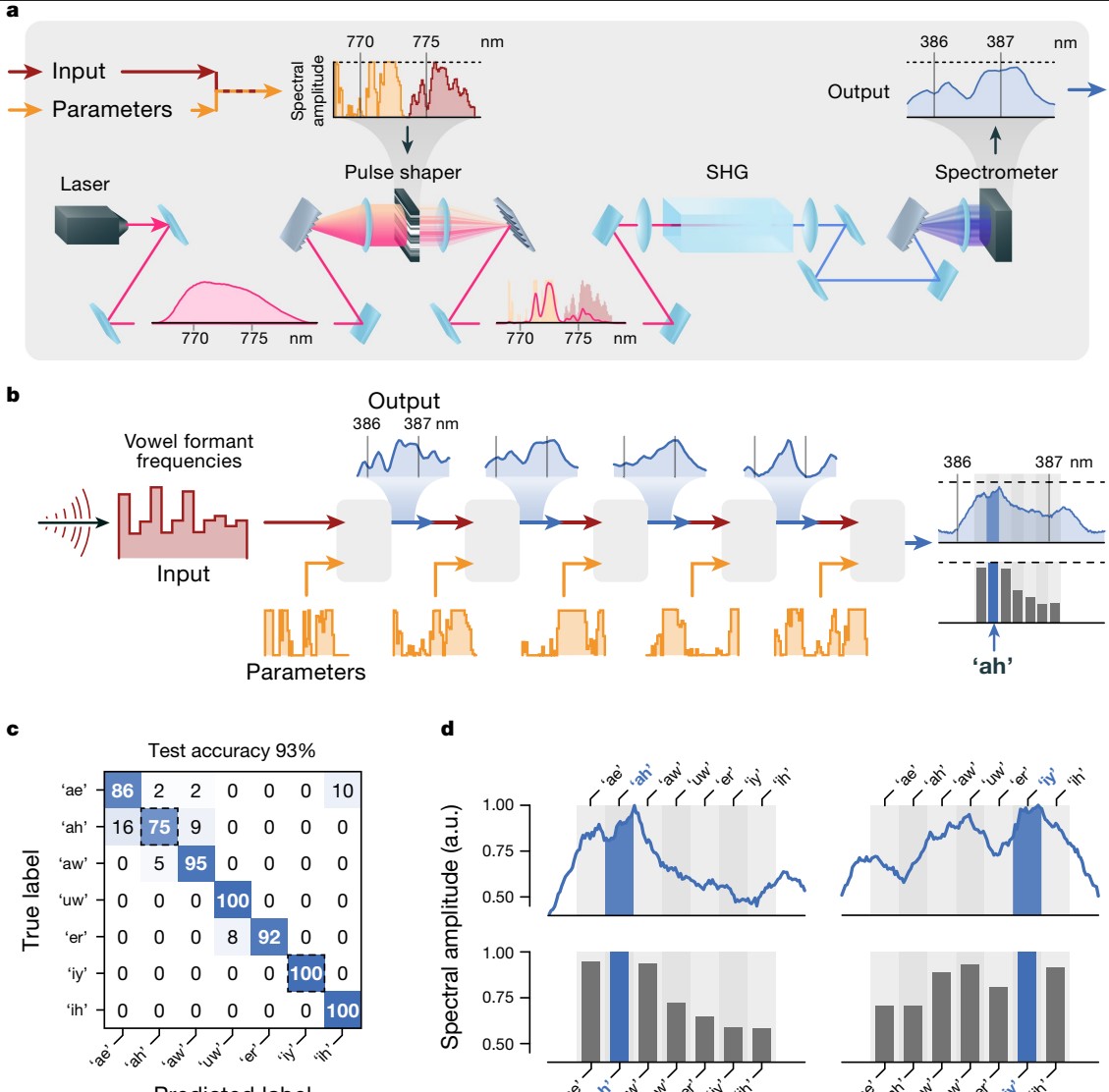

**Fig. 2 | An example PNN, implemented experimentally using broadband optical SHG. a**, Input data are encoded into the spectrum of a laser pulse (Methods, Supplementary Section 2). To control transformations implemented by the broadband SHG process, a portion of the pulse's spectrum is used as trainable parameters (orange). The physical computation result is obtained from the spectrum of a blue (about 390 nm) pulse generated within a $\chi^{(2)}$ medium. **b**, To construct a deep PNN, the outputs of the SHG transformations are used as inputs to subsequent SHG transformations, with independent trainable parameters. **c**, **d**, After training the SHG-PNN (see main text, Fig. 3), it classifies test vowels with 93% accuracy. **c**, The confusion matrix for the PNN on the test set. **d**, Representative examples of final-layer output spectra, which show the SHG-PNN's prediction.

classification using three distinct systems: the multimode mechanical oscillations of a driven metal plate, the analogue dynamics of a nonlinear electronicoscillator and ultrafast optical second-harmonic generation (SHG). We obtain accurate hierarchical classifiers that utilize each system's unique physical transformations, and that inherently mitigate each system's unique noise processes and imperfections. Although PNNs are a radical departure from traditional hardware, it is easy to integrate them into modern machine learning. We show that PNNs can be seamlessly combined with conventional hardware and neural network methods via physical–digital hybrid architectures, in which conventional hardware learns to opportunistically cooperate with unconventional physical resources using PAT. Ultimately, PNNs provide routes to improving the energy efficiency and speed of machine learning by many orders of magnitude, and pathways to automatically designing complex functional devices, such as functional nanoparticles[28], robots[25,26] and smart sensors[30–32].

## An example PNN based on nonlinear optics

Figure 2 shows an example PNN based on broadband optical pulse propagation in quadratic nonlinear media (ultrafast SHG). Ultrafast SHG realizes a physical computation roughly analogous to a nonlinear convolution, transforming the input pulse's near-infrared spectrum (about 800-nm centre wavelength) into the blue (about 400 nm) through a multitude of nonlinear frequency-mixing processes (Methods). To control this computation, input data and parameters are encoded into sections of the spectrum of the near-infrared pulse by modulating its frequency components using a pulse shaper (Fig. 2a). This pulse then propagates through a nonlinear crystal, producing a blue pulse whose spectrum is measured to read out the result of the physical computation.

To realize vowel classification with SHG, we construct a multilayer SHG-PNN (Fig. 2b) where the input data for the first physical layer consist of a vowel-formant frequency vector. After the final physical layer,

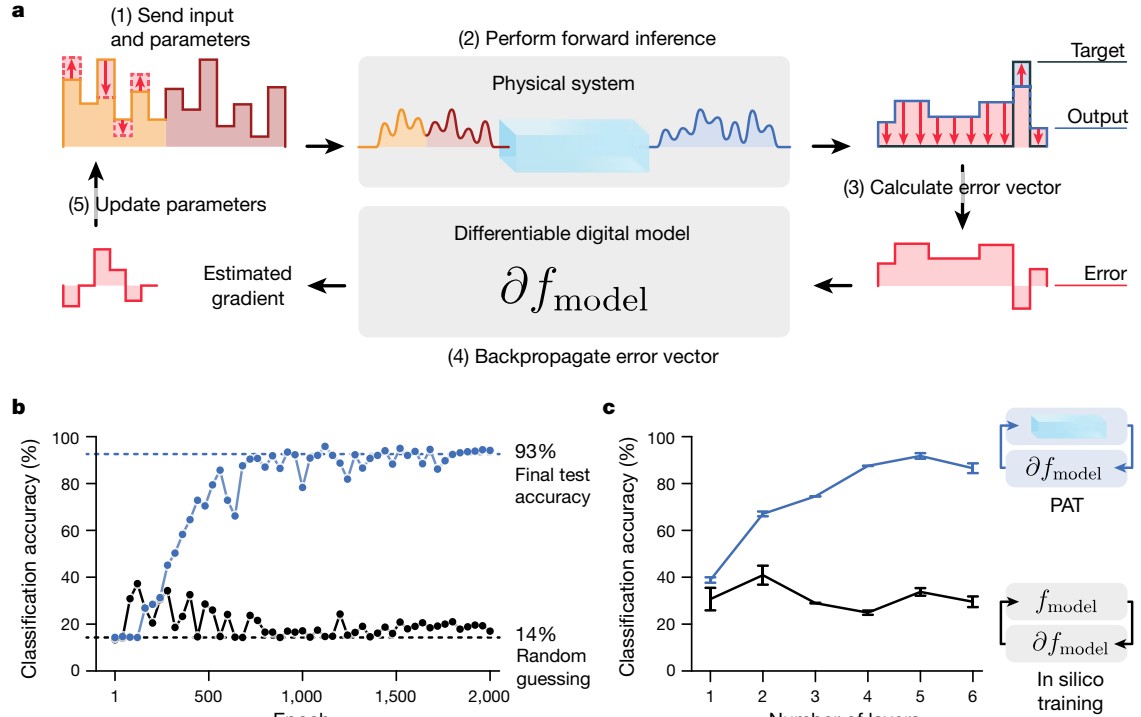

**Fig. 3 | Physics-aware training. a**, PAT is a hybrid in situ–in silico algorithm to apply backpropagation to train controllable physical parameters so that physical systems perform machine-learning tasks accurately even in the presence of modelling errors and physical noise. Instead of performing the training solely within a digital model (in silico), PAT uses the physical systems to compute forward passes. Although only one layer is depicted in **a**, PAT generalizes naturally to multiple layers (Methods). **b**, Comparison of the validation accuracy versus training epoch with PAT and in silico training, for the experimental SHG-PNN depicted in Fig. 2b. **c**, Final experimental test accuracy for PAT and in silico training for SHG-PNNs with increasing numbers of physical layers. The length of error bars represent two standard errors.

the blue output spectrum is summed using a digital computer into seven spectral bins (Fig. 2b, d, Supplementary Figs. 21, 22). The predicted vowel is identified by the bin with the maximum energy (Fig. 2c). In each layer, the output spectrum is digitally renormalized before being passed to the next layer (via the pulse shaper), along with a trainable digital rescaling. Mathematically, this transformation is given by $\mathbf{x}^{[l+1]} = \frac{a\mathbf{y}^{[l]}}{\max(\mathbf{y}^{[l]})} + b$, where $\mathbf{x}^{[l]}$ and $\mathbf{y}^{[l]}$ are the inputs and outputs of the $[l]$th layer, respectively, and $a$ and $b$ are scalar parameters of the transformation. Thus, the SHG-PNN's computations are carried out almost entirely by the trained optical transformations, without digital activation functions or output layers.

Deep PNNs essentially combine the computational philosophy of techniques such as PRC[21,22] with the trained hierarchical computations and gradient-based training of deep learning. In PRC, a physical system, often with recurrent dynamics, is used as an untrained feature map and a trained linear output layer (typically on a digital computer) combines these features to approximate desired functions. In PNNs, the backpropagation algorithm is used to adjust physical parameters so that a sequence of physical systems performs desired computations physically, without needing an output layer. For additional details, see Supplementary Section 3.

## Physics-aware training

To train the PNNs' parameters using backpropagation, we use PAT (Fig. 3). In the backpropagation algorithm, automatic differentiation determines the gradient of a loss function with respect to trainable parameters. This makes the algorithm $N$-times more efficient than finite-difference methods for gradient estimation (where $N$ is the number of parameters). The key component of PAT is the use of mismatched forward and backward passes in executing the backpropagation algorithm. This technique is well known in neuromorphic computing[48–53], appearing recently in direct feedback alignment[52] and quantization-aware training[48], which inspired PAT. PAT generalizes these strategies to encompass arbitrary physical layers, arbitrary physical network architectures and, more broadly, to differentially programmable physical devices.

PAT proceeds as follows (Fig. 3). First, training input data (for example, an image) are input to the physical system, along with trainable parameters. Second, in the forward pass, the physical system applies its transformation to produce an output. Third, the physical output is compared with the intended output to compute the error. Fourth, using a differentiable digital model, the gradient of the loss is estimated with respect to the controllable parameters. Finally, the parameters are updated according to the inferred gradient. This process is repeated, iterating over training examples, to reduce the error. See Methods for the intuition behind why PAT works and the general multilayer algorithm.

The essential advantages of PAT stem from the forward pass being executed by the actual physical hardware, rather than by a simulation. Our digital model for SHG is very accurate (Supplementary Fig. 20) and includes an accurate noise model (Supplementary Figs. 18, 19). However, as evidenced by Fig. 3b, in silico training with this model still fails, reaching a maximum vowel-classification accuracy of about 40%. In contrast, PAT succeeds, accurately training the SHG-PNN, even when additional layers are added (Fig. 3b, c).

## Diverse PNNs for image classification

PNNs can learn to accurately perform more complex tasks, can be realized with virtually any physical system and can be designed with a variety of physical network architectures. In Fig. 4, we present three PNN classifiers for the MNIST (Modified National Institute of Standards

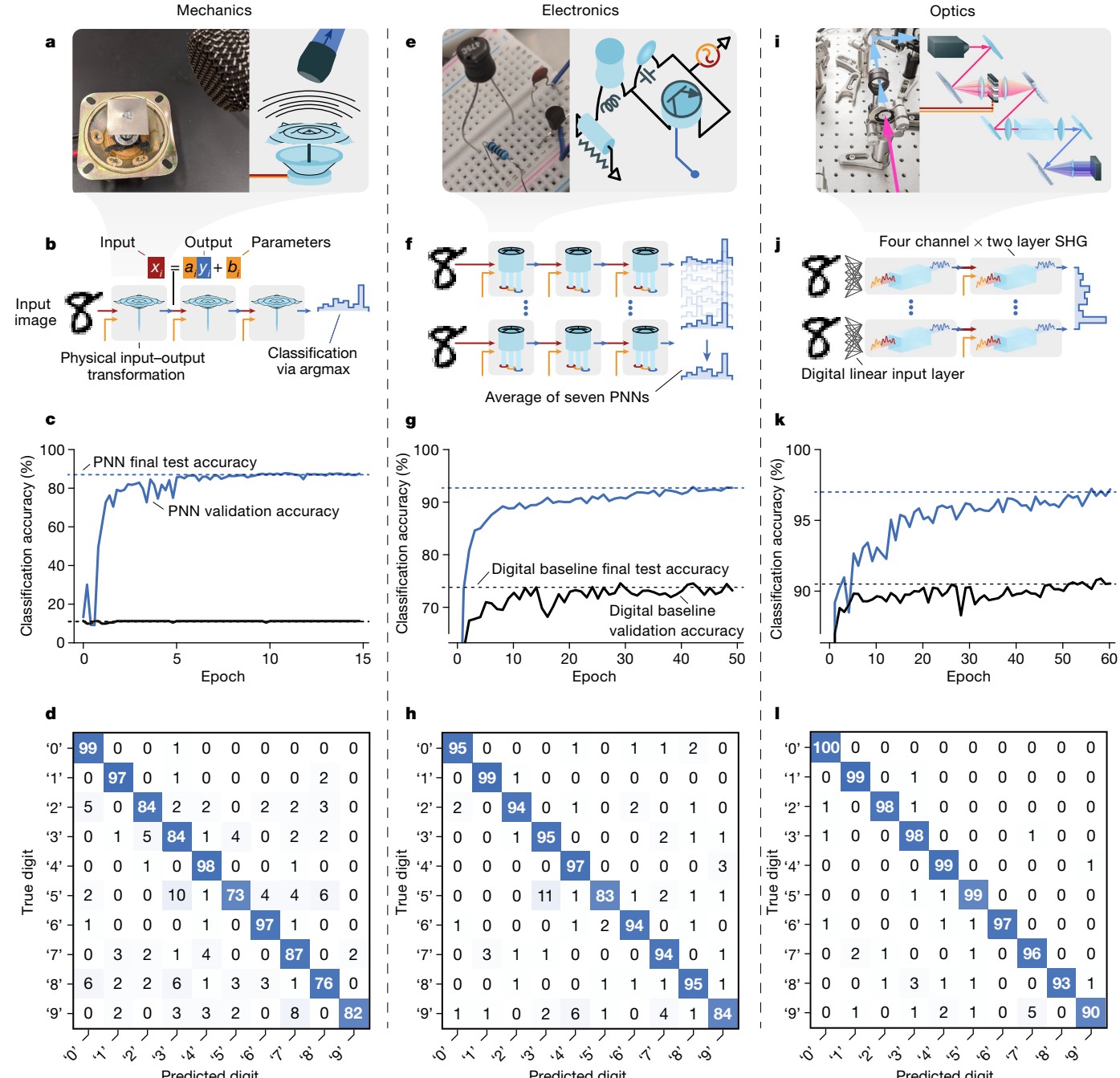

**Fig. 4 | Image classification with diverse physical systems.** We trained PNNs based on three physical systems (mechanics, electronics and optics) to classify images of handwritten digits. **a**, The mechanical PNN: the multimode oscillations of a metal plate are driven by time-dependent forces that encode the input image data and parameters. **b**, The mechanical PNN multilayer architecture. **c**, The validation classification accuracy versus training epoch for the mechanical PNN trained using PAT. The same curves are shown also for a reference model where the physical transformations implemented by the speaker are replaced by identity operations. **d**, Confusion matrix for the mechanical PNN after training. **e–h**, The same as **a–d**, respectively, but for a nonlinear analogue-electronic PNN. **i–l**, The same as **a–d**, respectively, for a hybrid physical–digital PNN based on broadband optical SHG. The final test accuracy is 87%, 93% and 97% for the mechanical, electronic and optics-based PNNs, respectively.

and Technology database) handwritten digit classification task, based on three distinct physical systems. For each physical system, we also demonstrate a different PNN architecture, illustrating the variety of physical networks possible. In all cases, models were constructed and trained using PyTorch[54].

In the mechanical PNN (Fig. 4a–d), a metal plate is driven by time-varying forces, which encode both input data and trainable parameters. The plate's multimode oscillations enact controllable

convolutions on the input data (Supplementary Figs. 16, 17). Using the plate's trainable transformation sequentially three times, we classify 28-by-28 (784 pixel) images that are input as an unrolled time series. To control the transformations of each physical layer, we train element-wise rescaling of the forces applied to the plate (Fig. 4b, Methods). PAT trains the three-layer mechanical PNN to 87% accuracy, close to a digital linear classifier[55]. When the mechanical computations are replaced by identity operations, and only the digital rescaling

operations are trained, the performance of the model is equivalent to random guessing (10%). This shows that most of the PNN's functionality comes from the controlled physical transformations.

An analogue-electronic PNN is implemented with a circuit featuring a transistor (Fig. 4e–h), which results in a noisy, nonlinear transient response (Supplementary Figs. 12, 13). The usage and architecture of the electronic PNN are mostly similar to that of the mechanical PNN, with several minor differences (Methods). When trained using PAT, the analogue-electronic PNN performs the classification task with 93% test accuracy.

Using broadband SHG, we demonstrate a physical–digital hybrid PNN (Fig. 4i–l). This hybrid PNN involves trainable digital linear input layers followed by trainable ultrafast SHG transformations. The trainable SHG transformations boost the performance of the digital baseline from roughly 90% accuracy to 97%. The classification task's difficulty is nonlinear with respect to accuracy, so this improvement typically requires increasing the number of digital operations by around one order of magnitude[55]. This illustrates how a hybrid physical–digital PNN can automatically learn to offload portions of a computation from an expensive digital processor to a fast, energy-efficient physical co-processor.

To show the potential for PNNs to perform more challenging tasks, we simulated a multilayer PNN based on a nonlinear oscillator network. This PNN is trained with PAT to perform the MNIST task with 99.1% accuracy, and the Fashion-MNIST task, which is considered significantly harder[56], with 90% accuracy, in both cases with simulated physical noise, and with mismatch between model and simulated experiment of over 20% (Supplementary Section 4).

## Discussion

Our results show that controllable physical systems can be trained to execute DNN calculations. Many systems that are not conventionally used for computation appear to offer, in principle, the capacity to perform parts of machine-learning-inference calculations orders of magnitude faster and more energy-efficiently than conventional hardware (Supplementary Section 5). However, there are two caveats to note. First, owing to underlying symmetries and other constraints, some systems may be well suited for accelerating a restricted class of computations that share the same constraints. Second, PNNs trained using PAT can only provide significant benefits during inference, as PAT uses a digital model. Thus, as in the hybrid network presented in Fig. 4i–l, we expect such PNNs to serve as a resource, rather than as a complete replacement, for conventional general-purpose hardware (Supplementary Section 5).

Techniques for training hardware in situ[7,40–47] and methods for reliable in silico training (for example, refs. [57–60]) complement these weaknesses. Devices trained using in situ learning algorithms will perform learning entirely in hardware, potentially realizing learning faster and more energy-efficiently than current approaches. Such devices are suited to settings in which frequent retraining is required. However, to perform both learning and inference, these devices have more specific hardware requirements than inference-only hardware, which may limit their achievable inference performance. In silico training can train many physical parameters of a device, including ones set permanently during fabrication[12–16]. As the resulting hardware will not perform learning, it can be optimized for inference. Although accurate, large-scale in silico training has been implemented[4–6,57–60], this has been achieved with only analogue electronics, for which accurate simulations and controlled fabrication processes are available. PAT may be used in settings where a simulation–reality gap cannot be avoided, such as if hardware is designed at the limit of fabrication tolerances, operated outside usual regimes or based on platforms other than conventional electronics.

Improvements to PAT could extend the utility of PNNs. For example, PAT's backward pass could be replaced by a neural network that directly estimates parameter updates for the physical system. Implementing this 'teacher' neural network with a PNN would allow subsequent training to be performed without digital assistance.

This work has focused so far on the potential application of PNNs as accelerators for machine learning, but PNNs are promising for other applications as well, particularly those in which physical, rather than digital, data are processed or produced. PNNs can perform computations on data within its physical domain, allowing for smart sensors[30–32] that pre-process information before conversion to the electronic domain (for example, a low-power, microphone-coupled circuit tuned to recognize specific hotwords). As the achievable sensitivity, resolution and energy efficiency of many sensors is limited by conversion of information to the digital electronic domain, and by processing of that data in digital electronics, PNN sensors should have advantages. More broadly, with PAT, one is simply training the complex functionality of physical systems. Although machine learning and sensing are important functionalities, they are but two of many[23–32] that PAT, and the concept of PNNs, could be applied to.

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

## Methods

### Physics-aware training

To train the PNNs presented in Figs. 2–4, we used PAT to enable us to perform backpropagation on the physical apparatuses as automatic differentiation (autodiff) functions within PyTorch[54] (v1.6). We used PyTorch Lightning[61] (v0.9) and Weights and Biases[62] (v0.10) during development as well. PAT is explained in detail in Supplementary Section 1, where it is compared with standard backpropagation, and training physical devices in silico. Here we provide only an overview of PAT in the context of a generic multilayer PNN (Supplementary Figs. 2, 3).

PAT can be formalized by the use of custom constituent autodiff functions for the physically executed submodules in an overall network architecture (Supplementary Fig. 1). In PAT, each physical system's forward functionality is provided by the system's own controllable physical transformation, which can be thought of as a parameterized function $f_p$ that relates the input $\mathbf{x}$, parameters $\boldsymbol{\theta}$, and outputs $\mathbf{y}$ of the transformation via $\mathbf{y} = f_p(\mathbf{x}, \boldsymbol{\theta})$. As a physical system cannot be auto-differentiated, we use a differentiable digital model $f_m$ to approximate each backward pass through a given physical module. This structure is essentially a generalization of quantization-aware training[48], in which low-precision neural network hardware is approximated by quantizing weights and activation values on the forward pass, but storing weights and activations, and performing the backward pass with full precision.

To see how this works, we consider here the specific case of a multilayer feedforward PNN with standard stochastic gradient descent. In this case, the PAT algorithm with the above-defined custom autodiff functions results in the following training loop:

Perform forward pass:

$$\mathbf{x}^{[l+1]} = \mathbf{y}^{[l]} = f_p(\mathbf{x}^{[l]}, \boldsymbol{\theta}^{[l]}) \tag{1}$$

Compute (exact) error vector:

$$g_{\mathbf{y}^{[M]}} = \frac{\partial L}{\partial \mathbf{y}^{[N]}} = \frac{\partial \ell}{\partial \mathbf{y}^{[N]}}(\mathbf{y}^{[N]}, \mathbf{y}_{\text{target}}) \tag{2}$$

Perform backward pass

$$g_{\mathbf{y}^{[l-1]}} = \left[ \frac{\partial f_m}{\partial \mathbf{x}}(\mathbf{x}^{[l]}, \boldsymbol{\theta}^{[l]}) \right]^{\text{T}} g_{\mathbf{y}^{[l]}} \tag{3a}$$

$$g_{\boldsymbol{\theta}^{[l-1]}} = \left[ \frac{\partial f_m}{\partial \boldsymbol{\theta}}(\mathbf{x}^{[l]}, \boldsymbol{\theta}^{[l]}) \right]^{\text{T}} g_{\mathbf{y}^{[l]}} \tag{3b}$$

Update parameters:

$$\boldsymbol{\theta}^{[l]} \rightarrow \boldsymbol{\theta}^{[l]} - \eta \frac{1}{N_{\text{data}}} \sum_k g_{\boldsymbol{\theta}^{[l]}}^{(k)} \tag{4}$$

where $g_{\boldsymbol{\theta}^{[l]}}$ and $g_{\mathbf{y}^{[l]}}$ are estimators of the physical systems' exact gradients, $\frac{\partial L}{\partial \boldsymbol{\theta}^{[l]}}$ and $\frac{\partial L}{\partial \mathbf{y}^{[l]}}$, respectively for the [l]th layer, obtained by auto-differentiation of the model, $L$ is the loss, $\ell$ is the loss function (for example, cross-entropy or mean-squared error), $\mathbf{y}_{\text{target}}$ is the desired (target) output, $N_{\text{data}}$ is the size of the batch and $\eta$ is the learning rate. $\mathbf{x}^{[l+1]}$ is the input vector to the [l + 1]th layer, which for the hidden layers of the feedforward architecture is equal to the output vector of the previous layer, $\mathbf{x}^{[l+1]} = \mathbf{y}^{[l]} = f_p(\mathbf{x}^{[l]}, \boldsymbol{\theta}^{[l]})$, where $\boldsymbol{\theta}^{[l]}$ is the controllable (trainable) parameter vector for the [l]th layer. For the first layer, the input data vector $\mathbf{x}^{[1]}$ is the data to be operated on. In PAT, the error vector is exactly estimated ($g_{\mathbf{y}^{[N]}} = \frac{\partial L}{\partial \mathbf{y}^{[N]}}$) as the forward pass is performed by the physical system. This error vector is then backpropagated via equation (3), which involves Jacobian matrices of the

differential digital model evaluated at the correct inputs at each layer (that is, the actual physical inputs) $\left[ \frac{\partial f_m}{\partial \mathbf{x}}(\mathbf{x}^{[l]}, \boldsymbol{\theta}^{[l]}) \right]^{\text{T}}$, where T represents the transpose operation. Thus, in addition to utilizing the output of the PNN ($\mathbf{y}^{[N]}$) via physical computations in the forward pass, intermediate outputs ($\mathbf{y}^{[l]}$) are also utilized to facilitate the computation of accurate gradients in PAT.

As it is implemented just by defining a custom autodiff function, generalizing PAT for more complex architectures, such as multichannel or hybrid physical–digital models, with different loss functions and so on is straightforward. See Supplementary Section 1 for details.

An intuitive motivation for why PAT works is that the training's optimization of parameters is always grounded in the true optimization landscape by the physical forward pass. With PAT, even if gradients are estimated only approximately, the true loss function is always precisely known. As long as the gradients estimated by the backward pass are reasonably accurate, optimization will proceed correctly. Although the required training time is expected to increase as the error in gradient estimation increases, in principle it is sufficient for the estimated gradient to be pointing closer to the direction of the true gradient than its opposite (that is, that the dot product of the estimated and true gradients is positive). Moreover, by using the physical system in the forward pass, the true output from each intermediate layer is also known, so gradients of intermediate physical layers are always computed with respect to correct inputs. In any form of in silico training, compounding errors build up through the imperfect simulation of each physical layer, leading to a rapidly diverging simulation–reality gap as training proceeds (see Supplementary Section 1 for details). As a secondary benefit, PAT ensures that learned models are inherently resilient to noise and other imperfections beyond a digital model, as the change of loss along noisy directions in parameter space will tend to average to zero. This makes training robust to, for example, device–device variations, and facilitates the learning of noise-resilient (and, more speculatively, noise-enhanced) models[8].

### Differentiable digital models

To perform PAT, a differentiable digital model of the physical system's input–output transformation is required. Any model, $f_m$, of the physical system's true forward function, $f_p$, can be used to perform PAT, so long as it can be auto-differentiated. Viable approaches include traditional physics models, black-box machine-learning models[13,63,64] and physics-informed machine-learning[65] models.

In this work, we used the black-box strategy for our differentiable digital models, namely DNNs trained on input–output vector pairs from the physical systems as $f_m$ (except for the mechanical system). Two advantages of this approach are that it is fully general (it can be applied even to systems in which one has no underlying knowledge-based model of the system) and that the accuracy can be extremely high, at least for physical inputs, $(\mathbf{x}, \boldsymbol{\theta})$, within the distribution of the training data (for out-of-distribution generalization, we expect physics-based approaches to offer advantages). In addition, the fact that each physical system has a precise corresponding DNN means that the resulting PNN can be analysed as a network of DNNs, which may be useful for explaining the PNN's learned physical algorithm.

For our DNN differentiable digital models, we used a neural architecture search[66] to optimize hyperparameters, including the learning rate, number of layers and number of hidden units in each layer. Typical optimal architectures involved 3–5 layers with 200–1,000 hidden units in each, trained using the Adam optimizer, mean-squared loss function and learning rates of around $10^{-4}$. For more details, see Supplementary Section 2D.1.

For the nonlinear optical system, the test accuracy of the trained digital model (Supplementary Fig. 20) shows that the model is remarkably accurate compared with typical simulation–experiment agreement in broadband nonlinear optics, especially considering that the pulses

used exhibit a complex spatiotemporal structure owing to the pulse shaper. The model is not, however, an exact description of the physical system: the typical error for each element of the output vector is about 1–2%. For the analogue electronic circuit, agreement is also good, although worse than the other systems (Supplementary Fig. 23), corresponding to around 5–10% prediction error for each component of the output vector. For the mechanical system, we found that a linear model was sufficient to obtain excellent agreement, which resulted in a typical error of about 1% for each component of the output vector (Supplementary Fig. 26).

## In silico training
To train PNNs in silico, we applied a training loop similar to the one described above for PAT except that both the forward and backward passes are performed using the model (Supplementary Figs. 1, 3), with one exception noted below.

To improve the performance of in silico training as much as possible and permit the fairest comparison with PAT, we also modelled the input-dependent noise of the physical system and used this within the forward pass of in silico training. To do this, we trained, for each physical system, an additional DNN to predict the eigenvectors of the output vector's noise covariance matrix, as a function of the physical system's input vector and parameter vector. These noise models thus provided an input- and parameter-dependent estimate of the distribution of noise in the output vector produced by the physical system. We were able to achieve excellent agreement between the noise models' predicted noise distributions and experimental measurements (Supplementary Figs. 18, 19). We found that including this noise model improved the performance of experiments performed using parameters derived from in silico training. Consequently, all in silico training results presented in this paper make use of such a model, except for the mechanical system, where a simpler, uniform noise model was found to be sufficient. For additional details, see Supplementary Section 2D.2.

Although including complex, accurate noise models does not allow in silico training to perform as well as PAT, we recommend that such models be used whenever in silico training is performed, such as for physical architecture search and design and possibly pre-training (Supplementary Section 5), as the correspondence with experiment (and, in particular, the predicted peak accuracy achievable there) is significantly improved over simpler noise models, or when ignoring physical noise.

## Ultrafast nonlinear optical pulse propagation experiments
For experiments with ultrafast nonlinear pulse propagation in quadratic nonlinear media (Supplementary Figs. 8–10), we shaped pulses from a mode-locked titanium:sapphire laser (Spectra Physics Tsunami, centred around 780 nm and pulse duration around 100 fs) using a custom pulse shaper. Our optical pulse shaper used a digital micromirror device (DMD, Vialux V-650L) and was inspired by the design in ref. [67]. Despite the binary modulations of the individual mirrors, we were able to achieve multilevel spectral amplitude modulation by varying the duty cycle of gratings written to the DMD along the dimension orthogonal to the diffraction of the pulse frequencies. To control the DMD, we adapted code developed for ref. [68], which is available at ref. [69].

After being shaped by the pulse shaper, the femtosecond pulses were focused into a 0.5-mm-long beta-barium borate crystal. The multitude of frequencies within the broadband pulses then undergo various nonlinear optical processes, including sum-frequency generation and SHG. The pulse shaper imparts a complex phase and spatiotemporal structure on the pulse, which depend on the input and parameters applied through the spectral modulations. These features would make it impossible to accurately model the experiment using a one-dimensional pulse propagation model. For simplicity, we refer to this complex, spatiotemporal quadratic nonlinear pulse propagation as ultrafast SHG.

Although the functionality of the SHG-PNN does not rely on a closed-form mathematical description or indeed on any form of mathematical isomorphism, some readers may find it helpful to understand the approximate form of the input–output transformation realized in this experimental apparatus. We emphasize that the following model is idealistic and meant to convey key intuitions about the physical transformation: the model does not describe the experimental transformation in a quantitative manner, owing to the numerous experimental complexities described above.

The physical transformation of the ultrafast SHG setup is seeded by the infrared light from the titanium:sapphire laser. This ultrashort pulse can be described by the Fourier transform of the electric field envelope of the pulse, $A_0(\omega)$, where $\omega$ is the frequency of the field detuned relative to the carrier frequency. For simplicity, consider a pulse consisting of a set of discrete frequencies or frequency bins, whose spectral amplitudes are described by the discrete vector $\mathbf{A_0} = [A_0(\omega_1), A_0(\omega_2), ..., A_0(\omega_N)]^T$. After passing through the pulse-shaper, the spectral amplitudes of the pulse are then given by

$$\mathbf{A} = [\sqrt{x_1}A_0(\omega_1), \sqrt{x_2}A_0(\omega_2), ..., \sqrt{\theta_1}A_0(\omega_{N_x+1}), \sqrt{\theta_2}A_0(\omega_{N_x+2}), ...]^T, \quad (5)$$

where $N_x$ is the dimensionality of the data vector, $\theta_i$ are the trainable pulse-shaper amplitudes and $x_i$ are the elements of the input data vector. Thus, the output from the pulse shaper encodes both the machine-learning data as well as the trainable parameters. Square roots are present in equation (5) because the pulse shaper was deliberately calibrated to perform an intensity modulation.

The output from the pulse shaper (equation (5)) is then input to the ultrafast SHG process. The propagation of an ultrashort pulse through a quadratic nonlinear medium results in an input–output transformation that roughly approximates an autocorrelation, or nonlinear convolution, assuming that the dispersion during propagation is small and the input pulse is well described by a single spatial mode. In this limit, the output blue spectrum $B(\omega_i)$ is mathematically given by

$$B(\omega_i) = k \sum_j A(\omega_i + \omega_j)A(\omega_i - \omega_j), \quad (6)$$

where the sum is over all frequency bins $j$ of the pulsed field. The output of the trainable physical transformation $\mathbf{y} = f_p(\mathbf{x}, \boldsymbol{\theta})$ is given by the blue pulse's spectral power, $\mathbf{y} = [|B_{\omega_1}|^2, |B_{\omega_2}|^2, ..., |B_{\omega_N}|^2]^T$, where $N$ is the length of the output vector.

From this description, it is clear that the physical transformation realized by the ultrafast SHG process is not isomorphic to any conventional neural network layer, even in this idealized limit. Nonetheless, the physical transformation retains some key features of typical neural network layers. First, the physical transformation is nonlinear as the SHG process involves the squaring of the input field. Second, as the terms within the summation in equation (6) involve both parameters and input data, the transformation also mixes the different elements of the input data and parameters to product an output. This mixing of input elements is similar, but not necessarily directly mathematically equivalent to, the mixing of input vector elements that occur in the matrix-vector multiplications or convolutions that appear in conventional neural networks.

## Vowel classification with ultrafast SHG
A task often used to demonstrate novel machine-learning hardware is the classification of spoken vowels according to formant frequencies[10,11]. The task involves predicting the spoken vowels given a 12-dimensional input data vector of formant frequencies extracted from audio recordings[10]. Here we use the vowel dataset from ref. [10], which is based on data originally from ref. [70]; data available at https://homepages.wmich.edu/~hillenbr/voweldata.html. This dataset consists of 273 data input–output pairs. We used 175 data pairs as the training

set−49 for the validation and 49 for the test set. For the results in Figs. 2, 3, we optimized for the hyperparameters of the PNN architecture using the validation error and only evaluated the test error after all optimization was conducted. In Fig. 3c, for each PNN with a given number of layers, the experiment was conducted with two different training, validation and test splits of the vowel data. In Fig. 3c, the line plots the mean over the two splits, and the error bars are the standard error of the mean.

For the vowel-classification PNN presented in Figs. 2, 3, the input vector to each SHG physical layer is encoded in a contiguous short-wavelength section of the spectral modulation vector sent to the pulse shaper, and the trainable parameters are encoded in the spectral modulations applied to the rest of the spectrum. For the physical layers after the first layer, the input vector to the physical system is the measured spectrum obtained from the previous layer. For convenience, we performed digital renormalization of these output vectors to maximize the dynamic range of the input and ensure that inputs were within the allowed range of 0 to 1 accepted by the pulse shaper. Relatedly, we found that training stability was improved by including additional trainable digital re-scaling parameters to the forward-fed vector, allowing the overall bias and amplitude scale of the physical inputs to each layer to be adjusted during training. These digital parameters appear to have a negligible role in the final trained PNN (when the physical transformations are replaced by identity operations, the network can be trained to perform no better than chance, and the final trained values of the scale and bias parameters are all very close to 1 and 0, respectively). We hypothesize that these trainable rescaling parameters are helpful during training to allow the network to escape noise-affected subspaces of parameter space. See Supplementary Section 2E.1 for details.

The vowel-classification SHG-PNN architecture (Supplementary Fig. 21) was designed to be as simple as possible while still demonstrating the use of a multilayer architecture with a physical transformation that is not isomorphic to a conventional DNN layer, and so that the computations involved in performing the classification were essentially all performed by the physical system itself. Many aspects of the design are not optimal with respect to performance, so design choices, such as our specific choice to partition input data and parameter vectors into the controllable parameters of the experiment, should not be interpreted as representing any systematic optimization. Similarly, the vowel-classification task was chosen as a simple example of multidimensional machine-learning classification. As this task can be solved almost perfectly by a linear model, it is in fact poorly suited to the nonlinear optical transformations of our SHG-PNN, which are fully nonlinear (Supplementary Figs. 9, 10). Overall, readers should not interpret this PNN's design as suggestive of optimal design strategies for PNNs. For initial guidelines on optimal design strategies, we instead refer readers to Supplementary Section 5.

### MNIST handwritten digit image classification with a hybrid physical–digital SHG-PNN

The design of the hybrid physical–digital MNIST PNN based on ultrafast SHG for handwritten digit classification (Fig. 4i–l) was chosen to demonstrate a proof-of-concept PNN in which substantial digital operations were co-trained with substantial physical transformations, and in which no digital output layer was used (although a digital output layer can be used with PNNs, and we expect such a layer will usually improve performance, we wanted to avoid confusing readers familiar with reservoir computing, and so avoided using digital output layers in this work).

The network (Supplementary Fig. 29) involves four trainable linear input layers that operate on MNIST digit images, whose outputs are fed into four separate channels in which the SHG physical transformation is used twice in succession (that is, it is two physical layers deep). The output of the final layers of each channel (the final SHG spectra) are concatenated, then summed into ten bins to perform a classification. The structure of the input layer was chosen to minimize the complexity of inputs to the pulse shaper. We found that the output second-harmonic spectra produced by the nonlinear optical process tended towards featureless triangular spectra if inputs were close to a random uniform distribution. Thus, to ensure that output spectra varied significantly with respect to changes in the input spectral modulations, we made sure that inputs to the pulse shaper would exhibit a smoother structure in the following way. For each of 4 independent channels, 196-dimensional input images (downsampled from 784-dimensional 28 × 28 images) are first operated on by a 196 by 50 trainable linear matrix, and then (without any nonlinear digital operations), a second 50 by 196 trainable linear matrix. The second 50 by 196 matrix is identical for all channels, the intent being that this matrix identifies optimal 'input modes' to the SHG process. By varying the middle dimension of this two-step linear input layer, one may control the amount of structure (number of 'spectral modes') allowed in inputs to the pulse shaper, as the middle dimension effectively controls the rank of the total linear matrix. We found that a middle dimension below 30 resulted in the most visually varied SHG output spectra, but that 50 was sufficient for good performance on the MNIST task. In this network, we also utilized skip connections between layers in each channel. This was done so that the network would be able to 'choose' to use the linear digital operations to perform the linear part of the classification task (for which nearly 90% accuracy can be obtained[55]) and to thus rely on the SHG co-processor primarily for the harder, nonlinear part of the classification task. Between the physical layers in each channel, a trainable, element-wise rescaling was used to allow us to train the second physical layer transformations efficiently. That is, $x_i = a_i y_i + b_i$, where $b_i$ and $a_i$ are trainable parameters, and $x_i$ and $y_i$ are the input to the pulse shaper and the measured output spectrum from the previous physical layer, respectively.

For further details on the nonlinear optical experimental setup and its characterization, we refer readers to Supplementary Section 2A. For further details on the vowel-classification SHG-PNN, we refer readers to Supplementary Section 2E.1, and for the hybrid physical–digital MNIST handwritten digit-classification SHG-PNN, we refer readers to Supplementary Section 2E.4.

### Analogue electronic circuit experiments

The electronic circuit used for our experiments (Supplementary Fig. 11) was a resistor-inductor-capacitor oscillator (RLC oscillator) with a transistor embedded within it. It was designed to produce as nonlinear and complex a response as possible, while still containing only a few simple components (Supplementary Figs. 12, 13). The experiments were carried out with standard bulk electronic components, a hobbyist circuit breadboard and a USB data acquisition (DAQ) device (Measurement Computing USB-1208-HS-4AO), which allowed for one analogue input and one analogue output channel, with a sampling rate of 1 MS s$^{-1}$.

The electronic circuit provides only a one-dimensional time-series input and one-dimensional time-series output. As a result, to partition the inputs to the system into trainable parameters and input data so that we could control the circuit's transformation of input data, we found it was most convenient to apply parameters to the one-dimensional input time-series vector by performing trainable, element-wise rescaling on the input time-series vector. That is, $x_i = a_i y_i + b_i$, where $b_i$ and $a_i$ are trainable parameters, $y_i$ are the components of the input data vector and $x_i$ are the re-scaled components of the voltage time series that is then sent to the analogue circuit. For the first layer, $y_i$ are the unrolled pixels of the input MNIST image. For hidden layers, $y_i$ are the components of the output voltage time-series vector from the previous layer.

We found that the electronic circuit's output was noisy, primarily owing to the timing jitter noise that resulted from operating the DAQ at its maximum sampling rate (Supplementary Fig. 23). Rather than reducing this noise by operating the device more slowly, we were motivated to design the PNN architecture presented in Fig. 4 in a way that

allowed it to automatically learn to function robustly and accurately, even in the presence of up to 20% noise per output vector element (See Supplementary Fig. 24 for an expanded depiction of the architecture). First, seven, three-layer feedforward PNNs were trained together, with the final prediction provided by averaging the output of all seven, three-layer PNNs. Second, skip connections similar to those used in residual neural networks were employed[71]. These measures make the output of the network effectively an ensemble average over many different subnetworks[71], which allows it to perform accurately and train smoothly despite the very high physical noise and multilayer design.

For further details on the analogue electronic experimental setup and its characterization, we refer readers to Supplementary Section 2B. For further details on the MNIST handwritten digit-classification analogue electronic PNN, we refer readers to Supplementary Section 2E.2.

### Oscillating mechanical plate experiments

The mechanical plate oscillator was constructed by attaching a 3.2 cm by 3.2 cm by 1 mm titanium plate to a long, centre-mounted screw, which was fixed to the voice coil of a commercial full-range speaker (Supplementary Figs. 14, 15). The speaker was driven by an audio amplifier (Kinter K2020A+) and the oscillations of the plate were recorded using a microphone (Audio-Technica ATR2100x-USB Cardioid Dynamic Microphone). The diaphragm of the speaker was completely removed so that the sound recorded by the microphone is produced only by the oscillating metal plate.

As the physical input (output) to (from) the mechanical oscillator is a one-dimensional time series, similar to the electronic circuit, we made use of element-wise trainable rescaling to conveniently allow us to train the oscillating plate's physical transformations.

The mechanical PNN architecture for the MNIST handwritten digit classification task was chosen to be the simplest multilayer PNN architecture possible with such a one-dimensional dynamical system (Supplementary Fig. 27). As the mechanical plate's input–output responses are primarily linear convolutions (Supplementary Figs. 16, 17), it is well suited to the MNIST handwritten digit classification task, achieving nearly the same performance as a digital linear model[55].

For further details on the oscillating mechanical plate experimental setup and its characterization, we refer readers to Supplementary Section 2C. For further details on the MNIST handwritten digit-classification oscillating mechanical plate PNN, we refer readers to Supplementary Section 2E.3.

### Data availability

All data generated during and code used for this work are available at https://doi.org/10.5281/zenodo.4719150.

### Code availability

An expandable demonstration code for applying PAT to train PNNs is available at https://github.com/mcmahon-lab/Physics-Aware-Training. All code used for this work is available at https://doi.org/10.5281/zenodo.4719150.

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

**Acknowledgements** We thank NTT Research for their financial and technical support. Portions of this work were supported by the National Science Foundation (award CCF-1918549). L.G.W. and T.W. acknowledge support from Mong Fellowships from Cornell Neurotech during early parts of this work. P.L.M. acknowledges membership of the CIFAR Quantum Information Science Program as an Azrieli Global Scholar. We acknowledge discussions with D. Ahsanullah, M. Anderson, V. Kremenetski, E. Ng, S. Popoff, S. Prabhu, M. Saebo, H. Tanaka, R. Yanagimoto, H. Zhen and members of the NTT PHI Lab/NSF Expeditions research collaboration, and thank P. Jordan for discussions and illustrations.

**Author contributions** L.G.W., T.O. and P.L.M. conceived the project and methods. T.O. and L.G.W. performed the SHG-PNN experiments. L.G.W. performed the electronic-PNN experiments. M.M.S. performed the oscillating-plate-PNN experiments. T.W., D.T.S. and Z.H. contributed to initial parts of the work. L.G.W., T.O., M.M.S. and P.L.M. wrote the manuscript. P.L.M. supervised the project.

**Competing interests** L.G.W., T.O., M.M.S. and P.L.M. are listed as inventors on a US provisional patent application (number 63/178,318) on physical neural networks and physics-aware training. The other authors declare no competing interests.

**Additional information**
**Correspondence and requests for materials** should be addressed to Logan G. Wright, Tatsuhiro Onodera or Peter L. McMahon.
