## [Peer Review File · Nature]

Manuscript Title: Deep physical neural networks trained with backpropagation

Reviewer Comments & Author Rebuttals

Reviewer Reports on the Initial Version:

Referee #1 (Remarks to the Author):

In this work, Wright et al propose to train physical systems to do tasks, using backpropagation with a new "physics-aware training". The main idea is, in this training, to use the actual experimental system for the forward pass and an approximate model of the system for the backward pass (which cannot be realized experimentally). This is a luminous idea that has the potential to inspire many. The paper does have some limitations -- I would have loved some more advanced tasks -- but the idea is verified experimentally in several experimental systems, the results are impressive, convincing, and well-controlled. The paper is written in a very thinking-forward way, which makes the reader think. I was very enthusiastic when reading the papers, and it gave me many ideas of variations of this approach to try in my own research. I feel that the paper is appropriate for publication in Nature, and I can see people trying physics-aware training to train systems even in other fields (chemistry, bioengineering...).

Mismatching the forward pass and the backward pass, and the fact that the backward pass does not need to match the trained system perfectly is a modern idea in deep learning, used especially for training low precision neural networks. So, the fact that the approach of the paper works will not surprise deep learning practitioners who have worked on this particular topic. On the other hand, it will be highly surprising for physicists and other machine learning experts, and I consider the paper very significant.

The paper gives plenty of technical details in Methods and Supplementary Information. The code is available open-source on github, allowing the reader to reproduce the approach.

Comments:

- In the discussion, I think that too much emphasis is put on the "self-simulation advantage". I think that the valid comparison is the advantage over the implementation of a similar task on conventional hardware (CPU, GPU, TPU...), which, in many situations, will be a lot less than the self-simulation advantage. For example, for MNIST, in a conventional implementation, you will use MLP to solve MNIST, which is simpler than the ordinary differential equations you would use to simulate the physical system.

- Overall, it took me too much effort to understand how the output is extracted from the PNN experiments. It became really clear when I reviewed the "comparison to reservoir computer" supplementary note (which is useful). The manuscript should be improved with regard to this.

I am still confused by this sentence: "This choice was made to avoid causing confusion for readers familiar with reservoir computing, not necessarily because performance without an output layer is superior." This needs more explanation.

- Why does the vowel example require that many epochs? This is confusing, as the MNIST

examples require more reasonable numbers of epochs.

- I appreciate the experimental examples on MNIST. Some suggestions, e.g. by simulation, on how PAT scales to more difficult tasks would have been appreciated, but I don't think that it is required for this paper.

- Some experiments use MNIST and some use downscaled MNIST, and the paper makes some comparison between the recognition rates of both approaches ("Since it is nonlinear, the electronic PNN outperforms the mechanical PNN"). Did you control that the downscaling could not explain what you are seeing?

- The paper should include a discussion of the limitations of the approach -- things that cannot work.

- I very much appreciated the comment that the best impact of PAT is not necessarily for accelerating deep learning, but for processing data in the natural physics of the devices (smart sensor). Showing an example of this would be an exciting addition to the paper.

- The formatting of the Supplementary Information is very uncomfortable to read (small font with tiny line spacing).

- I very much appreciated the open-source availability of the code on GitHub. This can very significantly enhance the impact of the paper. (I had no time to try the code myself).

Referee #2 (Remarks to the Author):

Wright et al. propose to use physical hardware to perform the inference phase of the neural networks. Any physical system can be considered a computation engine that performs a series of computations (like a neural network) according to the laws of physics and some controllable parameters inputted to the system externally. However, to perform the desired task at hand, the controllable (or tunable) parameters of physical hardware must be adequately trained before it is deployed as inference hardware. Furthermore, the details of the physical system may not be known to the full extent; hence, training these physical systems on a conventional digital system (so-called *in silico* by the authors) is a challenge.

The authors propose to use "physics-aware training" (PAT) to overcome this challenge. PAT uses the physical hardware (so-called *in situ* by authors) to perform the forward propagation of the signals and use the approximate model of the physical hardware to backpropagate the error signals and calculate gradients on a conventional digital system (so-called *in silico* by the authors). The authors claim that using the physical hardware for the forward propagation provides better convergence (trained results) than fully digital (*in silico*) training using the approximate model.

PAT may be an original idea, but I believe the paper misses some major articles related to 1) adequately training physical systems completely *in-situ* without backpropagation and 2) adequately training physical systems completely *in silico*. You can find my detailed comments

below, but I think the current manuscript is partial and at most complementary given these alternative approaches. Therefore, the manuscript needs to introduce these alternatives properly and justify their PAT approach (at least as an alternative) before it can be published in Nature. Assuming authors can discuss their approach comparatively (pros and cons) in the view of these alternative approaches, I believe PAT is an original idea worth publishing in Nature.

Detailed Comments

- In-situ training using equilibrium propagation.

The backpropagation algorithm is a way to calculate the gradients efficiently, whereas SGD, Adam, or any other gradient-based optimizer is used to train the neural networks. Therefore, assuming there are other ways to compute the gradients efficiently, the backpropagation algorithm is not necessarily required (in contrast to what the authors claim). Indeed, equilibrium propagation originally published by Scellier and Bengio does exactly this gradient computation step without using the backpropagation algorithm. A few references related to equilibrium propagation are listed below.

- <https://www.frontiersin.org/articles/10.3389/fncom.2017.00024/full>

- <https://arxiv.org/abs/2005.04168>

- <https://www.frontiersin.org/articles/10.3389/fnins.2021.633674/full>

In the above references, equilibrium propagation claimed to train physical systems completely in-situ with only performing the forward propagation in two phases (without using the backpropagation). Therefore, this work is very relevant and needs to be mentioned properly.

- There has been significant progress using analog resistive (memristor) crossbar arrays for the inference phase of the neural networks. Memristor arrays are only very briefly mentioned by authors. Below are some important papers that address several important concerns raised by the authors, such as the scalability, tolerance to noise processes, and successful in-silico training of these analog crossbar arrays before their deployment as inference hardware.

Successful in silico training related papers

- <https://ieeexplore.ieee.org/abstract/document/8993573>, 'The marriage of training and inference for scaled deep learning analog hardware'

- <https://ieeexplore.ieee.org/abstract/document/9472868>, 'Noise-Resilient DNN: Tolerating Noise in PCM-Based AI Accelerators via Noise-Aware Training'

- <https://www.frontiersin.org/articles/10.3389/fncom.2021.675741/full>, 'Toward Software-Equivalent Accuracy on Transformer-Based Deep Neural Networks With Analog Memory Devices'

In the above references, the so-called "hardware-aware-training" and "noise-aware-training" seem to be successful in training scaled networks deployed on physical analog hardware for inference workloads. Below two papers are the successful hardware demonstration of analog crossbar running inference task after trained in silico.

- https://vlsisymposium.org/files/press_kit/2021_VLSI_tip_sheet_EN_v6.pdf, two relevant papers below demonstrating near software equivalent accuracy on MNIST and CIFAR10 problems

- Narayanan, P., Burr, G. W., et al. "Fully on-chip MAC at 14nm enabled by accurate row-wise programming of PCM-based weights and parallel vector-transport in duration-format". 2021 Symposium on VLSI Technology, Digest of Technical Papers, T13-3

- Khaddam-Aljameh, R., Eleftheriou, E., et al. "A 14nm CMOS and PCM-based In-Memory Compute Core using an array of 300ps/LSB Linearized CCO-based ADCs and local digital processing". 2021 Symposium on VLSI Technology, Digest of Technical Papers, JFS2-5

A few other relevant work

- <https://ieeexplore.ieee.org/abstract/document/9458494>, 'A Flexible and Fast PyTorch Toolkit for Simulating Training and Inference on Analog Crossbar Arrays'

-<https://ieeexplore.ieee.org/abstract/document/9371973>, 'Unassisted True Analog Neural Network Training Chip'

In the view of these above papers (and in contrast to what authors argue), it seems there has been significant progress that minimizes the simulation-reality gap and mitigates device-to-device imperfections and various noise sources. Therefore, I believe PAT has to be discussed while taking these studies into consideration.

Other comments:

- PNN (compared to digital emulation) may provide speed and power benefits only for the inference workloads. Whereas PAT uses PNN only for the inference (forward) path, and the remaining error backpropagation and gradient computation is done on a conventional digital system using an approximate model of the PNN. It is very well known that 1) forward propagation, 2) the error backpropagation, and 3) gradient computation steps require roughly the same computation resources. Therefore PAT can only deliver about 1/3 improvement in speed and energy compared to fully digital (in silico) training. I think this is not clear from the statements made by the authors. PAT only delivers good training results in terms of accuracy. (PAT does not provide significant speed and energy savings).

- I understand why PAT may provide better training accuracy than fully in-silico training due to the discrepancy between the approximate model and the ground truth computation performed by the physical system. However, suppose PAT is really required for good training accuracy. In that case, each physical hardware (even though they may look identical to an external user) must require retraining independently due to the possible slight discrepancy in their physical parameters (or any other hardware variability). Therefore, if PAT is strictly needed, it is a significant (but also negative) conclusion that each PNN will require its own training. This is clearly not the case for reduced precision (quantized) digital hardware used for inference workloads. The training can be performed using the quantized model in the forward pass (as mentioned by the authors). Once the training is complete, the model can be deployed on all sorts of digital hardware, since for reduced precision hardware, there is no gap between the model and reality.

Author Rebuttals to Initial Comments:

We thank both referees for their comments and suggestions. Several of the suggestions allowed us to see the manuscript (and the scientific concepts) in new ways, and we are genuinely excited by how much the manuscript has improved, especially for the broad readership of Nature.

Before addressing each referee's comments point-by-point, we would first like to summarize the main changes that have been made to the manuscript and supplementary material.

Summary of changes:

1. The introduction and main results sections have been improved to help differentiate the PNNs discussed in this work from existing approaches to deep neural network accelerators, and from techniques like reservoir computing. The introduction now also better reflects the state-of-the-art results achieved in analog electronic DNN accelerators, including those using *in silico* training, which is important context for consideration of when and where PNNs of the kind described in this work could have real impact in the future.
2. To demonstrate deep PNNs and PAT scaling to a more complex task, we performed new simulations with a network of coupled oscillators for the Fashion-MNIST task, which is known to be substantially more challenging than the traditional digit-MNIST task.
3. We have completely rewritten the discussion section to:

- a. Clearly discuss the limitations and challenges facing unconventional PNNs and PAT, including the fact that PAT requires a digital processor and therefore provides no potential speed-up or energy-efficiency improvements for training, only for inference.
- b. Comment on the trade-offs and relationship between PAT, existing *in situ* learning algorithms, in particular equilibrium propagation, and *in silico* training.
- c. Examine application areas where we expect PAT-related approaches to be useful in the near- and far-future.

In addition to the revised manuscript and supplementary material, we have included a version of the manuscript where the major changes made are highlighted in red text. For the sake of making this document readable, this changes-in-red version only shows the *major text changes* in response to referees below, not reference-renumbering, changes in the acknowledgements, etc.

Point-by-point response

Referee #1

R1.1

In this work, Wright et al propose to train physical systems to do tasks, using backpropagation with a new “physics-aware training”. The main idea is, in this training, to use the actual experimental system for the forward pass and an approximate model of the system for the backward pass (which cannot be realized experimentally). This is a luminous idea that has the potential to inspire many. The paper does have some limitations -- I would have loved some more advanced tasks -- but the idea is verified experimentally in several experimental systems, the results are impressive, convincing, and well-controlled. The paper is written in a very thinking-forward way, which makes the reader think. I was very enthusiastic when reading the papers, and it gave me many ideas of variations of this approach to try in my own research. I feel that the paper is appropriate for publication in Nature, and I can see people trying physics-aware training to train systems even in other fields (chemistry, bioengineering...).

Mismatching the forward pass and the backward pass, and the fact that the backward pass does not need to match the trained system perfectly is a modern idea in deep learning, used especially for training low precision neural networks. So, the fact that the approach of the paper works will not surprise deep learning practitioners who have worked on this particular topic. On the other hand, it will be highly surprising for physicists and other machine learning experts, and I consider the paper very significant.

The paper gives plenty of technical details in Methods and Supplementary Information. The code is available open-source on github, allowing the reader to reproduce the approach.

R1.1: We greatly appreciate this warm, encouraging response.

The comment on how mismatching the forward and backward pass is well-known to machine learning practitioners, but not to most other readers, also helped us to make sense of some trends we have observed with readers of the original manuscript. In short (see Other Changes section for more details), we had been implicitly assuming readers had a rather specialized set of background knowledge, when in fact the work's interdisciplinarity (and Nature's broad readership) requires more conservative background knowledge assumptions.

R1.2

- In the discussion, I think that too much emphasis is put on the "self-simulation advantage". I think that the valid comparison is the advantage over the implementation of a similar task on conventional hardware (CPU, GPU, TPU...), which, in many situations, will be a lot less than the self-simulation advantage. For example, for MNIST, in a conventional implementation, you will use MLP to solve MNIST, which is simpler than the ordinary differential equations you would use to simulate the physical system.

R1.2: This is absolutely true, and we thank the referee for being so understanding about a section that, when we've read it after a couple months, could easily have been misleading. As a result, we have chosen to completely remove any discussion of self-simulation advantage from the main article.

The "self-simulation advantage" is a potentially misleading quantity, but we have found *simulation analysis* to be very insightful in understanding how different physical systems can be used for computation, since it reveals which types of calculations they may excel at, and how to do engineer them to do so efficiently. (This logic was partly inspired by the study of self-simulation advantage, or quantum supremacy, in the quantum computing literature). We expect it will be part of a design methodology that any researchers who wish to design their own PNNs will also find useful.

As a result, we have chosen to keep some of the material on simulation analysis in the supplementary material, as part of a section meant to help readers who intend to design their own PNNs based on unconventional physical systems. In that text, the quantity formerly called 'self-simulation advantage' is now called a 'self-simulation quotient', since we think calling it an 'advantage' invites overly-optimistic interpretations, while 'quotient' makes it clear it is only an engineering figure-of-merit.

Changes:

- Removed all discussion of self-simulation advantage in the main article.

Changes to the supplementary material:

- Reworked the supplementary section on simulation analysis into a section on 'Considerations for useful physical neural networks based on unconventional physical substrates', and there introduced the self-simulation quotient with the proper caveats/caution.

R1.3

- Overall, it took me too much effort to understand how the output is extracted from the PNN experiments. It became really clear when I reviewed the "comparison to reservoir computer" supplementary note (which is useful). The manuscript should be improved with regard to this.

I am still confused by this sentence: "This choice was made to avoid causing confusion for readers familiar with reservoir computing, not necessarily because performance without an output layer is superior. " This needs more explanation.

R1.3

To help improve the clarity of the explanation of how the outputs are obtained from the PNN, we edited the text in the section where the first PNN for vowel classification is introduced.

Given that Physical Reservoir Computing (PRC) is a major inspiration for our work, but differs from it in crucial ways, we decided to add a short paragraph after introducing the first PNN example to directly explain the difference between the two techniques, and to refer readers to the supplementary section comparing PRC and PNNs. Since many readers will already be familiar with PRC, we expect this direct comparison to be helpful for them to understand PNNs. We hope that this paragraph captures the essence that made the "Comparison to reservoir computer" supplementary note clear.

Changes:

- Modified the text explaining how the first example PNN works, including how the output is obtained.
- Added a paragraph in the main article directly contrasting physical reservoir computing and PNNs, and explaining our choice to not include trained digital output layers in this work.
- Since the point about including output layers in PNNs is now clarified earlier in the main text, we deleted the relevant lines from the methods section.

R1.4

- Why does the vowel example require that many epochs? This is confusing, as the MNIST examples require more reasonable numbers of epochs.

R1.4

The dataset used for the vowel classification task is very small: it only includes 273 examples (pairs of formant-frequency vector and true vowel labels), whereas MNIST includes 60,000 examples (pairs of images and digit labels).

If one considers how many examples the network has seen during training (that is, one multiplies #epoch * examples in the training set), it turns out the number is within an order of magnitude: 546,000 for vowel classification, versus around 3,000,000 for MNIST, and vowel classification is an easier task.

Given that this is confusing and might suggest to readers that PAT is incredibly inefficient, we decided to address this point in the main text.

Changes:

- Added a line in the paragraph introducing the vowel classification task to explain this.

R1.5

- I appreciate the experimental examples on MNIST. Some suggestions, e.g. by simulation, on how PAT scales to more difficult tasks would have been appreciated, but I don't think that it is required for this paper.

R1.5

To address this, we decided to perform new simulations, as suggested by the Referee, for a PNN based on a network of oscillators, on the Fashion-MNIST task. Fashion MNIST is an established benchmark which is known to be significantly harder than conventional MNIST (see reference points below).

We chose to do simulations for this in part because we realized this would be an excellent example for our open-source demo code, and would allow us to demonstrate a PNN based on a system that should be familiar to a broader set of readers (in contrast, we have found the SHG system is intuitive to readers with an optics background, but is sometimes confusing for other readers).

To summarize the new simulations and results:

- **We simulated a PNN based on a network of nonlinearly-coupled nonlinear oscillators.** We included realistic physical noise and mismatch (i.e., simulation-reality gap) between the model used for PAT's backward pass and the (simulated) physical device.
- **Using PAT, we train a PNN based on 2 physical layers of oscillator networks to perform the Fashion MNIST task with 90% accuracy.** The Fashion MNIST task is known to be significantly more challenging than the original MNIST task; this performance is comparable to convolutional neural networks requiring several million multiplication operations per inference.
- **Using PAT, we can train the same 2-layer PNN to perform the original MNIST task with nearly state-of-the-art 99.1% accuracy.**
- **For both tasks, PAT trains the PNN to achieve the accuracy stated above even when the mismatch between the backward-pass model and physical device is roughly 20%.**
- **PAT results in significantly improved accuracy compared to *in silico* training.** With the same 20% model mismatch, *in silico* training results in much worse performance, similar to a linear digital model (which requires about 10^4 operations per inference).

Additional details and figures are provided below. For a complete description, see Section 4 of the revised Supplementary Material.

The system:

The network of oscillators is described by the following equations of motion:

$$\frac{d^2 q_i}{dt^2} = -\sin(q_i) + \sum_{j=1}^N J_{ij} [\sin(q_i) - \sin(q_j)] + e_i \quad (\text{Eqn. 1})$$

where q_i are the oscillator amplitudes, J_{ij} are the (symmetric) coupling coefficients, and e_i are the external drives for each oscillator.

These equations of motion are frequently approximated to derive the Frenkel-Kontorova model, which is a standard model in condensed-matter and nonlinear physics, with well-known applications to phonon physics and nonlinear wave propagation. Similar equations describe coupled lasers or pendula.

In training, the coupling coefficients J_{ij} are the trainable parameters. These coupling coefficients are not time-dependent. The input data is encoded as the initial oscillator amplitudes at $t = 0$, and the output is read from the oscillator amplitudes at a later time, $t = T$.

In order to simulate the effect of the mismatch between model and experiment that is addressed by physics-aware training, we add noise to the nonlinearity coefficient, and the coupling coefficients J_{ij} . Thus, while the model remains described by the equation above, the physical system is described by:

$$\frac{d^2 q_i}{dt^2} = -\sin([1 + \eta]q_i) + \sum_{j=1}^N [J_{ij} + J_{ij}^{noise}] [\sin([1 + \eta]q_i) - \sin([1 + \eta]q_j)] + e_i \quad (\text{Eqn. 2})$$

Finally, we consider physical noise in the device by randomly perturbing the initial amplitudes of the oscillators every time the equations are simulated. For all tests considered here, this noise is around 2% of the observed mean initial-time oscillator amplitude, $|q_i(t = 0)|$, which is around unity. **Thus, the physical system consists of simulating the system Eqn. 2 with this additive initial noise.**

When *in silico* training is performed, the model consists of Eqn. 1, and the noise is modelled by initial-condition fluctuations identical to those described in the above paragraph.

Fashion MNIST results:

The Fashion MNIST task is significantly more challenging than the original MNIST task. As reference points:

- Models that achieve 90% performance on Fashion MNIST typically can achieve ~99% on the original MNIST task. For a full comparison table, see <https://github.com/zalandoresearch/fashion-mnist>.
- A typical multilayer convolutional neural network (CNN) that achieves 90.3% test accuracy takes about 3 million multiplication operations (approximately 6 million total operations). This is roughly 10 times as many operations are required for LeNet-4, a multilayer CNN which achieves 99% test accuracy on the original MNIST digit task.
- The absolute maximum accuracy any classifier achieves on Fashion MNIST is 95.3%.

For the Fashion-MNIST task, we considered a small oscillator network in which all-to-all coupling is possible. We note that such all-to-all connectivity is not possible in every physical system, but for networks of oscillators connected by a bus, optical or RF electronic networks, or electronic circuit crossbar systems, such connectivity is achievable. We also made this choice to keep the example simple, since imposing connectivity constraints in a physically-accurate way requires significantly more complicated code than a first example should ideally include. Our PNN consists of a small 100-oscillator network that is intended to crudely mimic a convolutional layer, and a larger, 1610-oscillator network that serves as a kind of output layer.

We find that we can train a PNN based on two physical layers of coupled oscillators to perform the Fashion MNIST task with 90% test accuracy. This same PNN can also be trained to perform the original MNIST task with 99.1% accuracy, which is nearly state-of-the-art performance. To simulate the effect of a model mismatch that would occur when using physics-aware training, we assumed that the oscillator nonlinear rate and coupling strengths in the 'physical system' differed relative to the model used for backpropagation by 20% and roughly 30% respectively. Specifically, in Eqn. 2 $\eta = 0.2$, and J_{ij}^{noise} drawn from a normal distribution with standard deviation 0.3. Typical trained values of J_{ij} are about 0.25 (mean) or 1.0 (RMS), so this corresponds to a model mismatch of at least 20% overall.

Even with this large model mismatch, physics-aware training still trains the system adequately (Figure 1a). When *in silico* training is used instead, the PNN achieves 84% accuracy. While this may seem close to 90% accuracy in absolute terms, 84% is comparable performance to a single-layer perceptron requiring $784 \times 10 \sim 10^4$ multiplications to execute. Meanwhile, achieving the 90% accuracy is roughly equivalent to CNNs requiring $\sim 10^6$ operations to execute, so the difference in performance is significant.

When the model mismatch is increased further, to roughly 30% ($\eta = 0.3$ and J_{ij}^{noise} drawn from a normal distribution with standard deviation 0.5), PAT still performs relatively well, while *in silico* training degrades further (Figure 1b). With even larger model mismatch (about 40%, $\eta = 0.4$ and J_{ij}^{noise} drawn from a normal distribution with standard deviation 0.7), PAT's performance degrades slightly, but much less than the performance degradation that occurs for *in silico* training.

To summarize: PAT can train the PNN to perform this relatively complex task accurately even in the high model mismatch regime. The difference in performance between PAT and *in silico* training is large. When we increase the model mismatch further, we find that even as the performance of *in silico* training degrades, PAT still results in relatively good performance.

Figure 1: Physics-aware training compared to *in silico* training of a PNN based on a network of coupled nonlinear oscillators for the Fashion MNIST task. a. Validation accuracy during training for a model mismatch about 20%. **b.** Same, for a model mismatch of about 30%. **c.** Same, for a model mismatch of about 40%.

Although our new simulations are just one example, there are additional simulation results in the literature that give us confidence that PNNs will be able to perform complex machine learning tasks. For example, in Ref. 14 (Hughes et al., *Science Advances* (2019)), the authors use a very different system, nonlinear optical wave propagation in a structured waveguide, to perform the audio-domain version of the vowel classification task (which is also much harder than the original MNIST task) with over 90% accuracy. Of course, these results were obtained only in simulation, but this shows that the physical evolution is at least capable of performing a difficult machine learning task. We expect that if a similar experiment was performed with a reconfigurable device, it could be trained using PAT to achieve similar performance.

Overall, there is more to explore and understand regarding which physical systems make good PNNs for different tasks, as well as what is required for scalable PNNs, but we hope that we have shown strong evidence that unconventional PNNs and PAT do indeed have the potential to scale to more difficult tasks.

Changes:

- In the main article, we added a short paragraph prior to the discussion to introduce these results and refer readers to the Supplementary Section 4 where they are discussed.

Changes to the supplementary:

- Added a new section, Section 4, to the supplementary material describing these simulations.

R1.5

- Some experiments use MNIST and some use downsampled MNIST, and the paper makes some comparison between the recognition rates of both approaches ("Since it is nonlinear, the electronic PNN outperforms the mechanical PNN"). Did you control that the downscaling could not explain what you are seeing?

R1.5

This is a good point, since the task's difficulty does depend on the degree of downsampling. Fortunately, this hardness actually implies the difference between the two in this case PNNs is probably larger, rather than smaller.

The PNNs based on ultrafast second harmonic generation and nonlinear analog electronics both performed the MNIST task with downsampled images, which is indeed harder. In contrast, the mechanical PNN performed the MNIST task with full-resolution images, which is easier.

When we perform the MNIST task with the mechanical PNN with downsampled images, it performs significantly worse – test accuracy lower by 4% compared to the result with full-resolution images, reflecting the higher difficulty of the downsampled task.

That said, there are numerous differences between the mechanical PNN and electronic PNN besides nonlinearity, so we can see how this line probably encourages readers to wonder why analog electronics is better than mechanics, when in fact the comparison is not one-to-one in our work.

Changes:

- Removed the line in question, and replaced it with one that simply states the performance.

R1.6

- The paper should include a discussion of the limitations of the approach -- things that cannot work.

R1.6

This is an excellent suggestion, and we found it very easy to rewrite the Discussion section around it. We feel the new Discussion is still forward-thinking and cautiously optimistic, but hope it is equally open about the limitations and challenges of PAT and unconventional PNNs as it is about their potential positive impacts.

Briefly, the big limitations are:

1. **PAT, at least in its current form, still requires a digital computer during training.** To obtain a benefit, one needs to use the trained device in inference mode many times. In the future, it might be possible to train a second PNN to implement the part of PAT that is presently done with a digital computer.

2. **Even though PAT allows many new systems to be used for deep neural network calculations, most physical systems will provide large speed-ups/energy benefits only for a subclass of operations (e.g., if the physical system has some constraint or symmetry, the operations it can do efficiently will have a corresponding constraint).** This means that, in order to solve practical tasks, one will usually either needs to combine different physical systems together or augment the physical systems with some digital or otherwise flexible hardware.
3. **PAT relies on having a model for the system, and the model needs to be able to predict the system at least approximately.** Future improvements to PAT might be able to simplify this, such as training a neural network to predict the physical system's gradients, but for now, PAT is not model-free.

In order to also manage the length of the Discussion section, we have partly combined this discussion of limitations (and their possible improvements) with the discussion on how PAT relates to in-hardware learning techniques like equilibrium propagation, and *in silico* training methods, as suggested by Referee 2.

We now discuss limitations 1-2 in the first two paragraphs of the discussion, and limitation 3 in the third.

Last, we have also combined previous supplementary Sections 4 and 5 into a new section which is meant to help facilitate the design of new PNNs. This section includes a systematic discussion on the limitations of unconventional PNNs, and general guidelines that we expect will be necessary for PNNs to be useful.

Changes:

- Focused the discussion section, especially paragraphs 1 and 3, on pointing out limitations of our approach.

Changes to the Supplementary:

- Reworked the Supplementary Sections 4 and 5 as a set of rough guidelines and principles for PNNs that could become useful.

R1.7

- I very much appreciated the comment that the best impact of PAT is not necessarily for accelerating deep learning, but for processing data in the natural physics of the devices (smart sensor). Showing an example of this would be an exciting addition to the paper.

R1.7

Thank you for this – we agree. We have begun developing PNN smart sensors, and have quickly realized that this endeavor will require its own systematic paper. Currently, we have two projects underway to realize PNN smart sensors:

1. An on-chip phononic circuit PNN that will natively process microwave electronic signals (and possibly ultrasound signals) and,
2. An optical PNN smart sensor based on nonlinear optics in free space.

Since PNN sensors are themselves a vast new topic, and because this future paper will include several different authors from the present one, we feel it is best to leave out such an example in the current work.

We agree that applications like smart sensing are likely the most promising beneficiaries of PAT and unconventional PNNs in the near future, whereas the application of PNNs as machine-learning accelerators will probably take more time (in part because current analog electronic accelerators will likely need to reach fundamental limits before the benefit of PNNs would be significant). We think that, by including the discussion of these applications of PNNs in the very last paragraph of the article, it places appropriate emphasis on them and will, we hope, inspire readers to think of diverse applications of smart-sensor PNNs.

R1.8

- The formatting of the Supplementary Information is very uncomfortable to read (small font with tiny line spacing).

R1.8

Thank you for this feedback.

Changes:

- Increased the font size and line spacing in the supplementary material.

R1.9

- I very much appreciated the open-source availability of the code on GitHub. This can very significantly enhance the impact of the paper. (I had no time to try the code myself).

R1.9

Thank you for the encouragement – this comment also motivated us to develop the simulation described in R1.2 as an example for the demo code on Github, and we think it will be the most useful example included there.

Referee #2

Wright et al. propose to use physical hardware to perform the inference phase of the neural networks. Any physical system can be considered a computation engine that performs a

series of computations (like a neural network) according to the laws of physics and some controllable parameters inputted to the system externally. However, to perform the desired task at hand, the controllable (or tunable) parameters of physical hardware must be adequately trained before it is deployed as inference hardware. Furthermore, the details of the physical system may not be known to the full extent; hence, training these physical systems on a conventional digital system (so-called *in silico* by the authors) is a challenge.

The authors propose to use “physics-aware training” (PAT) to overcome this challenge. PAT uses the physical hardware (so-called *in situ* by authors) to perform the forward propagation of the signals and use the approximate model of the physical hardware to backpropagate the error signals and calculate gradients on a conventional digital system (so-called *in silico* by the authors). The authors claim that using the physical hardware for the forward propagation provides better convergence (trained results) than fully digital (*in silico*) training using the approximate model.

R2.1

PAT may be an original idea, but I believe the paper misses some major articles related to 1) adequately training physical systems completely *in-situ* without backpropagation and 2) adequately training physical systems completely *in silico*. You can find my detailed comments below, but I think the current manuscript is partial and at most complementary given these alternative approaches. Therefore, the manuscript needs to introduce these alternatives properly and justify their PAT approach (at least as an alternative) before it can be published in Nature. Assuming authors can discuss their approach comparatively (pros and cons) in the view of these alternative approaches, I believe PAT is an original idea worth publishing in Nature.

R2.1

We agree the literature pointed out by the Referee is very relevant. We have incorporated the Referee’s suggestions in the Introduction and Discussion sections, and we believe with these additions PAT and the unconventional deep PNNs we consider are better framed in the full context of existing techniques.

We will describe more specific changes in our responses below.

R2.2

- *In-situ* training using equilibrium propagation.

The backpropagation algorithm is a way to calculate the gradients efficiently, whereas SGD, Adam, or any other gradient-based optimizer is used to train the neural networks. Therefore, assuming there are other ways to compute the gradients efficiently, the backpropagation algorithm is not necessarily required (in contrast to what the authors claim). Indeed, equilibrium propagation originally published by Scellier and Bengio does exactly this gradient computation step without using the backpropagation algorithm. A few references related to equilibrium propagation are listed below.

- <https://www.frontiersin.org/articles/10.3389/fncom.2017.00024/full>

- <https://arxiv.org/abs/2005.04168>

- <https://www.frontiersin.org/articles/10.3389/fnins.2021.633674/full>

In the above references, equilibrium propagation claimed to train physical systems completely in-situ with only performing the forward propagation in two phases (without using the backpropagation). Therefore, this work is very relevant and needs to be mentioned properly.

R2.2

We agree that devices based on equilibrium propagation should be mentioned, especially because we find their pros/cons nicely complement those of devices based on physics-aware training, as well as those of devices trained *in silico*.

The Referee also noted an important point that backpropagation isn't required to perform gradient descent, and rather that only a method to infer gradients, or gradient-based updates, is required. In addition to being important, we appreciate this point because thinking about this comment inspired some suggestions we've made in the new discussion section for possible future improvements or alternatives to PAT, which we are excited about (see second-last paragraph of Discussion).

We have chosen to mainly address equilibrium propagation in the Discussion section, which we have completely rewritten. The Discussion section now focuses on the relative strengths and weaknesses of PAT compared to *in situ* learning techniques like equilibrium propagation, and to *in silico* training methods like the ones mentioned by the Referee in the next comment (R2.3). Overall, we believe the methods can work together quite synergistically, with each one's strengths and weaknesses complemented nicely by the others'.

In addition, we have now integrated a discussion of equilibrium propagation into the introduction. We have cited the references suggested by the Referee, along with a few others on the same topic we found helpful.

Changes:

- Modified the introduction and added lines contextualizing equilibrium propagation.
- Modified the introduction to note that physical learning with performance similar to backpropagation doesn't need the exact backpropagation algorithm per se, only the ability to estimate local physical gradients (and, of course, to update the parameters).
- Inspired by this comment, in the discussion section, added comments on how PAT might be improved on with methods to estimate (possibly physically) gradient-based updates.
- In the discussion section, we now contrast the limitations and plausible future of devices trained by PAT with those trained using equilibrium propagation and *in silico* training methods.

R2.3

- There has been significant progress using analog resistive (memristor) crossbar arrays for the inference phase of the neural networks. Memristor arrays are only very briefly mentioned by authors. Below are some important papers that address several important concerns raised by the authors, such as the scalability, tolerance to noise processes, and successful in-silico training of these analog crossbar arrays before their deployment as inference

hardware.

Successful *in silico* training related papers

-<https://ieeexplore.ieee.org/abstract/document/8993573>, 'The marriage of training and inference for scaled deep learning analog hardware'

-<https://ieeexplore.ieee.org/abstract/document/9472868>, 'Noise-Resilient DNN: Tolerating Noise in PCM-Based AI Accelerators via Noise-Aware Training'

-<https://www.frontiersin.org/articles/10.3389/fncom.2021.675741/full>, 'Toward Software-Equivalent Accuracy on Transformer-Based Deep Neural Networks With Analog Memory Devices'

In the above references, the so-called "hardware-aware-training" and "noise-aware-training" seem to be successful in training scaled networks deployed on physical analog hardware for inference workloads. Below two papers are the successful hardware demonstration of analog crossbar running inference task after trained *in silico*.

- https://vlsisymposium.org/files/press_kit/2021_VLSI_tip_sheet_EN_v6.pdf, two relevant papers below demonstrating near software equivalent accuracy on MNIST and CIFAR10 problems

- Narayanan, P., Burr, G. W., et al. "Fully on-chip MAC at 14nm enabled by accurate row-wise programming of PCM-based weights and parallel vector-transport in duration-format". 2021 Symposium on VLSI Technology, Digest of Technical Papers, T13-3

- Khaddam-Aljameh, R., Eleftheriou, E., et al. "A 14nm CMOS and PCM-based In-Memory Compute Core using an array of 300ps/LSB Linearized CCO-based ADCs and local digital processing". 2021 Symposium on VLSI Technology, Digest of Technical Papers, JFS2-5

A few other relevant work

-<https://ieeexplore.ieee.org/abstract/document/9458494>, 'A Flexible and Fast PyTorch Toolkit for Simulating Training and Inference on Analog Crossbar Arrays'

-<https://ieeexplore.ieee.org/abstract/document/9371973>, 'Unassisted True Analog Neural Network Training Chip'

In the view of these above papers (and in contrast to what authors argue), it seems there has been significant progress that minimizes the simulation-reality gap and mitigates device-to-device imperfections and various noise sources. Therefore, I believe PAT has to be discussed while taking these studies into consideration.

R2.3

Yes, this is a great point.

It is important context that *in silico* training can be made to work with novel devices and large-scale neural network models. The systems described in these papers are however based on relatively mature technologies, for which very detailed, accurate simulations exist, and for which fabrication and operation can be relatively precisely controlled.

We believe that PAT will be invaluable for scientific research and development of novel machine-learning hardware, but that its role in mature machine-learning hardware is, while promising, less certain. Regardless, we think that PAT will be valuable for the scientific community since alternative platforms like photonics and spintronics hold potential for

performing beyond the limits of conventional electronics. Alternatively, it is also possible that, in order to push any platform towards its physical limits, it may be beneficial to operate devices in regimes where fabrication is less controllable, and/or where the device physics is no longer accurately simulable with *ab initio* models. In such contexts, PAT may be helpful even if it is used only once, after device fabrication, to correct for the simulation-reality gap and recover the performance expected from simulations.

We have modified the introduction and discussion section to present the context of devices based on analog crossbar arrays and successful *in silico* training, and to contrast them with unconventional PNNs and PAT. In the discussion section, we have further addressed this point, where we comment on the relationship between *in silico* training, PAT, and *in situ* learning algorithms. We have cited the references suggested by the Referee, along with a few others on the same topic we found helpful.

Changes:

- The achievements of devices based on crossbar array analog electronics are now mentioned in the introduction where DNN accelerators are first introduced. We now contrast with these devices in the second paragraph by delineating between DNN accelerators based on a strict, or direct mathematical analogy to conventional DNNs, and those based on open, or more loose analogies, and also between DNN accelerators based on relatively-mature platforms like analog electronics with those based on much more unconventional systems like nonlinear photonics.
- We now comment on the successful achievements of *in silico* training in these platforms in the introduction paragraph where *in silico* training is first introduced. Here, we make the points above about why this may be challenging to achieve in other, less mature platforms.

R2.4

- PNN (compared to digital emulation) may provide speed and power benefits only for the inference workloads. Whereas PAT uses PNN only for the inference (forward) path, and the remaining error backpropagation and gradient computation is done on a conventional digital system using an approximate model of the PNN. It is very well known that 1) forward propagation, 2) the error backpropagation, and 3) gradient computation steps require roughly the same computation resources. Therefore PAT can only deliver about 1/3 improvement in speed and energy compared to fully digital (*in silico*) training. I think this is not clear from the statements made by the authors. PAT only delivers good training results in terms of accuracy. (PAT does not provide significant speed and energy savings).

R2.4

We agree that it is absolutely true that PAT does not provide any significant energy or speed benefit for training. The aspect of neural networks that PNNs trained using PAT can deliver speed or energy benefits to is in inference. For many applications (although certainly not all), a model is trained once and then used in inference mode a vast number of times, and in these applications the inference cost dominates. As some specific examples we could find: Ref. 1 reports that inference is responsible for 90% of energy consumption of machine learning at Amazon, and NVIDIA estimates a similar fraction, between 80 and 90%. We are

also optimistic that future neuromorphic devices could consist of 'frozen' parts that have been pretrained, much like the pretrained Transformer models that are gaining popularity in the deep-learning community today. In this case, hardware would absolutely need to be able to retrain itself to learn new tasks or adapt to new data, but some parts of the hardware could be left fixed all or most of the time. But regardless, even though solving the inference problem is important, we agree it is still only part of the larger problem. Training will always be essential, and the costs of training are already limiting research in the field of machine learning. We therefore strongly agree that an important accomplishment will be the acceleration of training by physics-based algorithms and specially-designed hardware.

We also think PAT will also be very useful for developing physical machine learning devices that operate directly on, or produce, data in the physical domain, such as smart sensors. For these applications, the cost to train could be well-worth paying, since these sensors would be able to process information in ways that traditional sensors cannot, and might enable scientific experiments that would otherwise be impossible. In many plausible settings, it will be beneficial to design smart sensors with relatively-unconventional physical substrates, such as nonlinear optical, microfluidic or mechanical devices, so PAT may be the only technique that is viable.

Changes:

- In the Discussion section, we explicitly point out that PAT does not offer a speed or energy benefit during training, and contrast this with methods like equilibrium propagation.
- In the Discussion section, we propose settings where devices trained with PAT will make sense given this limitation, as mentioned above.
- In the Discussion section, we suggest possible routes to improving PAT that might overcome this limitation, inspired by the Referee's previous comments about estimating gradient updates.

R2.5

- I understand why PAT may provide better training accuracy than fully in-silico training due to the discrepancy between the approximate model and the ground truth computation performed by the physical system. However, suppose PAT is really required for good training accuracy. In that case, each physical hardware (even though they may look identical to an external user) must require retraining independently due to the possible slight discrepancy in their physical parameters (or any other hardware variability). Therefore, if PAT is strictly needed, it is a significant (but also negative) conclusion that each PNN will require its own training. This is clearly not the case for reduced precision (quantized) digital hardware used for inference workloads. The training can be performed using the quantized model in the forward pass (as mentioned by the authors). Once the training is complete, the model can be deployed on all sorts of digital hardware, since for reduced precision hardware, there is no gap between the model and reality.

R2.5

Yes, this is true, and the Referee is absolutely correct that this restricts where we expect PNNs trained with PAT to be useful, and motivates future improvements to PAT that could

facilitate more physics-based learning, as well as large-scale experimental demonstrations of physics-based learning algorithms in general.

That said, we feel PAT is still a major step for several reasons. Some of our reasoning overlaps with comments made above, in responses R2.3 and R2.4. **In short: models derived using PAT seem to transfer very well between physical devices, even when there are large differences in physical parameters.** Thus, we expect in many scenarios PAT will not need to be used on each device individually, or will just be needed to perform a quick fine-tuning. But even if there are scenarios where each unique device needs to be individually trained, we expect that this individual device training will often be a worthy trade-off given the benefits and unique capabilities of PAT and devices trained with it. These scenarios include, for example:

- Early-stage research and development of new computing platforms (where the number of devices is ~ 1).
- Devices in which signal processing needs to take place in a physical domain that isn't normally used for computation, such as optical, microfluidic, or chemical 'smart sensors'. In some cases, PAT may be the only viable technique to train these devices, and many such scientific instruments could be one-of-a-kind.
- Execution (inference-only) of frozen parts of large machine learning models that exhibit good transfer learning, such as large-scale language models. PAT would be used to make frozen parts of the model very efficient, while other hardware capable of efficient learning would be used to implement retrainable parts, e.g., the last few layers of a deep network. Since the trained PNN would be used many times over the device's lifetime to execute inference of those frozen layers, using PAT to fine-tune after fabrication could be justified given the net performance gain.
- Inference on hardware that uses low-cost materials and fabrication methods, or that is otherwise designed at the limits of fabrication tolerances to maximize performance. Again, assuming the trained model is used many times after training, a one-time PAT fine-tuning could often be justified by the improvement in overall device cost, energy efficiency and speed.

Our investigations suggest that, in some contexts, it is not required to retrain devices from scratch, since we observe good transfer learning of models obtained with PAT on one device to a different, similar device. To illustrate this transfer learning, we have taken the Fashion-MNIST PNN described in R1.5, and considered what happens when the physical parameters learned by applying PAT to one physical device are transferred to a second physical device which has artificial 'fabrication errors' relative to the first device.

Here is a summary of these new simulations and the results:

- **We simulated a PNN based on networks of nonlinearly-coupled nonlinear oscillators.** We considered realistic physical noise and a 20% mismatch (i.e., simulation-reality gap) between the model used for PAT's backward pass and the (simulated) physical device.
- **We first use PAT to train a PNN based on 2 layers of oscillator networks to perform Fashion MNIST 90% accuracy.**
- **We create a second device with simulated fabrication errors by taking the first device and randomly perturbing its physical parameters by 6%.**
- **We then take the parameters learned using PAT on the original device, and directly transfer them to the second device. We find that the model transfers**

well: for 6% device-device variation, the second device still achieves 89.1% accuracy.

- If we apply PAT to train the second device after this parameter transfer, the original device's 90% accuracy is re-obtained (and much less training time is required than if the device was trained without this initialization).

More detailed description (See Supplementary Section 4 for complete details, and note that this description overlaps with R1.5):

Our simulated PNN is modelled by the following system of equations:

$$\frac{d^2 q_i}{dt^2} = -\sin(q_i) + \sum_{j=1}^N J_{ij} [\sin(q_i) - \sin(q_j)] + e_i \quad (\text{Eqn. 1})$$

where q_i are the oscillator amplitudes, J_{ij} are the (symmetric) coupling coefficients, and e_i are external drives for each oscillator.

In training, the coupling coefficients J_{ij} are the trainable parameters. These coupling coefficients are not time-dependent. The input data is encoded as the initial oscillator amplitudes at $t = 0$, and the output is read from the oscillator amplitudes at a later time, $t = T$.

To model the mismatch (i.e., simulation-reality gap) between model and physical device that PAT compensates for, we use the system of equations above as the digital model, and a simulation of a modified set of equations to stand in for a physical device being used as a PNN. Specifically, we modify the nonlinear coefficients and add offset noise to the coupling coefficients from Eqn. 1:

$$\frac{d^2 q_i}{dt^2} = -\sin([1 + \eta]q_i) + \sum_{j=1}^N [J_{ij} + J_{ij}^{noise}] [\sin([1 + \eta]q_i) - \sin([1 + \eta]q_j)] + e_i \quad (\text{Eqn. 2})$$

where J_{ij}^{noise} are drawn from a mean-zero normal distribution with standard deviation σ .

We first use PAT to train this first device to perform the Fashion MNIST task. We then transfer the J_{ij} parameters learned by PAT to a second device with additional artificial physical variation.

To model the variation between the first device and second device, we use a further-modified system of equations to describe the second device. For the second device, we modify the physical parameters by 30% of the model-device mismatch, beyond those in Eqn. 2. Concretely, the physical parameters are modified from the first device as: $\eta' = \eta + 0.3\eta$ and $J_{ij}^{noise'} = J_{ij}^{noise} + J_{ij}^{noise,new}$, where $J_{ij}^{noise,new}$ are drawn from a mean-zero normal distribution with standard deviation 0.3σ . The second device is thus described by:

$$\frac{d^2 q_i}{dt^2} = -\sin([1 + \eta']q_i) + \sum_{j=1}^N [J_{ij} + J_{ij}^{noise'}] [\sin([1 + \eta']q_i) - \sin([1 + \eta']q_j)] + e_i \quad (\text{Eqn. 3})$$

As an example, if the model mismatch is 20%, the device-device variation we consider is about 6% (i.e., 30% of 20%).

We also consider physical noise by randomly perturbing the initial conditions of the oscillators each time Eqn. 1, 2, or 3 is simulated, by about 2% of the mean initial-time oscillator amplitude.

Results of transfer of PAT-derived parameters between different devices:

Below, in Figure 2, the first confusion matrix (a) shows the performance of the first PNN when trained using *in silico* training, and the second confusion matrix (b) shows the performance of the PNN when trained using PAT. Finally, (c) shows the performance of the second PNN with the parameters J_{ij} transferred from PAT on the first PNN. The second PNN achieves 89.1% accuracy with these parameters, which suggests that the parameters derived using PAT are resilient to fabrication errors between devices.

Figure 2d-f shows the same sequence, but for increased device-model mismatch and device-device variation.

We note that the results shown in Figure 2 are without any retraining of the second device – if we initialize the second device with the model learned using PAT on the first device, and then apply PAT from that starting point, we can recover the performance of the first device, but with a much shorter training time than would have been required without the initialization.

To summarize: We find models obtained with PAT directly transfer well between devices with moderate variations in physical parameters (up to 9%). Though this is just a numerical model, we are overall optimistic that PAT could be used to train one device, once, and then the learned physical parameters could be used on many other devices of the same kind (but with uncontrolled variations), either directly, or as a starting point for a short PAT-based fine-tuning.

Figure 2: Confusion matrices showing the classified label predicted by the oscillator PNN versus the correct result for the Fashion MNIST task, for different training strategies and after transferring learned parameters between different devices. **a,b**, Confusions matrices for *in silico* training and physics aware training for a 20% mismatch between model and physical device. **c**, Confusion matrix when the parameters derived from PAT on a first device are transferred over to a second oscillator PNN with variations of about 6% from the first device. **d-f**. The same as a-c, but for 30% mismatch between model and device and 9% variation between devices.

Changes:

- Included in the Discussion section a paragraph addressing the contexts in which PAT may be useful relative to *in silico* training and *in situ* physical learning hardware. In short, we've emphasized that PAT is for inference-only applications, and also

emphasized the scientific applications of PAT to smart sensors and other settings where one needs to process data in an unconventional physical domain.

Changes to the supplementary material:

- We reworked previous sections to create a section (Section 5) discussing guidelines for PNNs and PAT, where some of these trade-offs are discussed in more detail.
- Added a figure in the new supplementary section, Section 4, showing simulations for the fashion MNIST task, where we demonstrate transfer learning of PAT models between different devices.

Other changes:

Since submitting the original manuscript, we have had colleagues read the manuscript and offer feedback, and we have noticed some room for improvement ourselves.

These changes are well-aligned with those suggested by the Referees, but in some cases do not overlap perfectly with any one comment. Rather than trying to shoehorn these changes into the direct responses above, we describe them here.

O1.1

First, while both reviewers understood this distinction very clearly, we noticed some readers found the original manuscript confusing regarding the distinction between physical DNN accelerators based on a direct, precise analogy, and those mainly discussed in this paper based on a much looser, more open analogy.

O1.1

To help improve this, we have introduced a delineation in the introduction between ‘strict analogy’ DNN accelerators, and the ‘more open analogy’ behind DNN accelerators we are discussing here. We have also tried to delineate between relatively mature platforms like analog electronics and unconventional platforms.

O1.2

Second, some of our colleagues were either completely unaware of methods in machine learning that utilize different forward and backward passes in backpropagation, or they knew a lot about these methods and felt that we did not sufficiently point out how well-known this basic trick is.

O1.2

In part inspired by Reviewer 1’s framing that the key insight behind PAT should not surprise modern ML researchers (which we completely agree with and had naively assumed readers would recognize), we added a few sentences after introducing PAT to contextualize it, both

for readers from machine learning who will recognize that the basic idea of mismatched forward and backward passes is well-known, and for readers from other fields like physics who may not have ever heard of techniques like quantization-aware training or direct feedback alignment.

O1.3

Last, and probably most critically, we found that many test readers were confused about precisely how deep physical neural networks differ from reservoir computers (which they were mostly familiar with).

Reviewer 1 also commented that the section directly comparing RC and PNNs was helpful.

O1.3

To address this, we have included a paragraph after introducing the first PNN for vowel classification, in which we explicitly compare PRC to PNNs trained with PAT, and refer the reader to the supplementary material section on this for more details.

Other small changes:

- We now cite PyTorch Lightning and Weights and Biases, which are software used in our experiments and simulations.

Reviewer Reports on the First Revision:

Referee #1 (Remarks to the Author):

The authors have very well taken into account my comments as well as those of the second reviewer. The new manuscript feels very solid. The new Discussion is deeper and overall vastly superior to the first version. I am enthusiastic to recommend the publication of the manuscript!

The new comparison of PNNs with reservoir computers (L216) makes me a little uncomfortable, as it ignores the recurrent and dynamical nature of reservoir computers, which allows them to deal with time-dependent tasks. This paragraph needs to be reworked. This ambivalence is also present in Supplementary Note 3, although it had not shocked me on my first read.

Also, I do not think that PNNs and reservoir computers have a "nature-as-computer philosophy." PNNs and reservoir computers are still artificial systems, they are not more natural than conventional computers.

In the new Discussion, I would replace the expression "unconventional physical systems" with something more precise.

Damien Querlioz

Referee #2 (Remarks to the Author):

The authors addressed all the issues raised by myself in the first review. Now they properly reference and discuss approaches that are alternatives to PAT. These changes make the paper much more solid and balanced. I mentioned in my first review that PAT is a unique idea worth publishing in Nature, and with these changes in place, I have no hesitation in endorsing the paper for its publication in Nature.

A minor suggestion in case the authors feel it is relevant. In lines 55 and 56, where the authors mention training innovations, I believe the following two papers might be appropriate to add along with Ref. 7

1) "Enabling Training of Neural Networks on Noisy Hardware"
"<https://www.frontiersin.org/articles/10.3389/frai.2021.699148/full>

2) "Algorithm for Training Neural Networks on Resistive Device Arrays"
<https://www.frontiersin.org/articles/10.3389/fnins.2020.00103/full>

Author Rebuttals to First Revision:

Referee #1

R1.1

The new comparison of PNNs with reservoir computers (L216) makes me a little uncomfortable, as it ignores the recurrent and dynamical nature of reservoir computers, which allows them to deal with time-dependent tasks. This paragraph needs to be reworked. This ambivalence is also present in Supplementary Note 3, although it had not shocked me on my first read.

Also, I do not think that PNNs and reservoir computers have a "nature-as-computer philosophy." PNNs and reservoir computers are still artificial systems, they are not more natural than conventional computers.

R1.1: We appreciate this comment and have reworked the paragraph to include that PRC often utilizes recurrent dynamics. We also agree that the term "nature-as-computer" does not carry the right connotations.

Changes:

We have modified the text to make clear that PRCs typically involve recurrent dynamics, and have removed the phrasing "nature-as-computer".

Modified Text:

Deep physical neural networks essentially combine the computational philosophy of techniques like physical reservoir computing (PRC)²⁰⁻²¹ with the trained hierarchical computations and gradient-based training of deep learning. In PRC, a physical system, often with recurrent dynamics, is used as an untrained feature map and a trained linear output layer (typically on a digital computer) combines these features to approximate desired functions. In PNNs, the backpropagation algorithm is used to adjust physical parameters so that a sequence of physical evolutions performs desired computations physically, without need for an output layer. For additional details, see Supplementary Section 3.

R1.2

In the new Discussion, I would replace the expression "unconventional physical systems" with something more precise.

R1.2: Yes, this is not as precise as it should be. We have modified the text to qualify what "unconventional" refers to.

Changes:

In the Discussion, we rephrased this to make it clear we mean systems that are not conventionally used for computation.

Modified Text:

Our results show that controllable physical systems can be trained to execute deep-neural-network calculations. Many systems that are not conventionally used for computation appear to offer, in principle, the capacity to perform parts of machine-learning-inference calculations orders of magnitude faster, and more energy-efficiently than conventional hardware (Supplementary Section 5). However, there are two caveats to note. First, due to underlying symmetries and other constraints, some systems may be well-suited only to accelerating a restricted class of computations that share the same constraints. Second, PNNs trained using PAT can only provide significant benefits during inference, since PAT uses a digital model. Thus, as in the hybrid network presented in Fig. 4i-l, we expect such PNNs to serve as a resource, rather than a complete replacement, for conventional general-purpose hardware (see Supplementary Section 5).

Referee #2

R2.1

A minor suggestion in case the authors feel it is relevant. In lines 55 and 56, where the authors mention training innovations, I believe the following two papers might be appropriate to add along with Ref. 7

1) "Enabling Training of Neural Networks on Noisy Hardware"
["https://www.frontiersin.org/articles/10.3389/frai.2021.699148/full"](https://www.frontiersin.org/articles/10.3389/frai.2021.699148/full)

2) "Algorithm for Training Neural Networks on Resistive Device Arrays" <https://www.frontiersin.org/articles/10.3389/fnins.2020.00103/full>

R2.1: We appreciate the suggestions as these references also show algorithms that may be used by analog crossbar array hardware to learn *in situ*, and they are different from any of the other algorithms already cited.

Changes:

Due to the reference limits in the main text, we added these references to the Supplementary Material.